# Perivascular localization of macrophages in the intestinal mucosa is regulated by Nr4a1 and the microbiome

Masaki Honda[1,2,4], Bas G.J. Surewaard[1,4], Mayuki Watanabe[1], Catherine C. Hedrick[3], Woo-Yong Lee[1], Kirsty Brown[1], Kathy D. McCoy[1] & Paul Kubes [1✉]

While the ontogeny and recruitment of the intestinal monocyte/macrophage lineage has been studied extensively, their precise localization and function has been overlooked. Here we show by imaging the murine small and large intestines in steady-state that intestinal CX3CR1+ macrophages form an interdigitated network intimately adherent to the entire mucosal lamina propria vasculature. The macrophages form contacts with each other, which are disrupted in the absence of microbiome, monocyte recruitment ($Ccr2^{-/-}$), or monocyte conversion ($Nr4a1^{-/-}$). In dysbiosis, gaps exist between the perivascular macrophages correlating with increased bacterial translocation from the lamina propria into the blood-stream. The recruitment of monocytes and conversion to macrophages during intestinal injury is also dependent upon CCR2, Nr4a1 and the microbiome. These findings demonstrate a relationship between microbiome and the maturation of lamina propria perivascular macrophages into a tight anatomical barrier that might function to prevent bacterial translocation. These cells are also critical for emergency vascular repair.

[1] Department of Physiology and Pharmacology, Calvin, Phoebe and Joan Snyder Institute for Chronic Diseases, Cumming School of Medicine, University of Calgary, Calgary, AB T2N 4N1, Canada. [2] Department of Transplantation and Pediatric Surgery, Kumamoto University, 1-1-1 Honjo, Chuo-ku, Kumamoto 860−8556, Japan. [3] Division of Inflammation Biology, La Jolla Institute for Allergy and Immunology, La Jolla, CA 92037, USA. [4] These authors contributed equally: Masaki Honda, Bas G. J. Surewaard. ✉email: pkubes@ucalgary.ca

Every tissue has its own cadre of macrophages with specific phenotypes playing critical roles in local homeostasis, but also in mitigation of infections and tissue repair[1,2]. Each organ is initially seeded by the same yolk sac and fetal liver precursors and these cells then are likely imprinted in the local environment to become very specialized macrophages that self-renew within the tissue[3–5]. While this is absolutely the case for microglia in the brain, alveolar macrophage in lung and Kupffer cells in the liver, the macrophages within the intestine may be somewhat exceptional to this rule. The majority but likely not all intestinal macrophages are continuously replenished by bone marrow derived classical Ly6C$^{high (hi)}$ monocytes that enter the intestine via the CCR2-CCL2 axis[6,7]. A role for alternative Ly6C$^{low (lo)}$ (CX3CR1$^{hi}$CCR2$^{lo}$) monocytes is less compelling. Some studies report fewer mature macrophages in $Cx3cr1^{-/-}$ mice, suggesting that CX3CR1$^{hi}$Ly6C$^{lo}$ monocytes may also contribute to the macrophage population in the intestine[2]. However, adoptive transfer of CX3CR1$^{hi}$Ly6C$^{lo}$ monocytes results in sequestration to the vasculature with no entry into healthy intestine suggesting a lesser role for these cells[8,9].

An alternative scenario could be that CCR2$^+$ monocytes enter the intestinal lamina propria and there they convert to CX3CR1$^{hi}$CCR2$^{lo}$ monocytes that ultimately give rise to CX3CR1$^+$ mature macrophages. Indeed, there is a growing body of evidence that a monocyte to macrophage continuum exists in the lamina propria known as the waterfall effect[8,10]. Recent work has identified $Nr4a1$ as a master regulator of CCR2$^+$ monocyte conversion to CX3CR1$^+$ monocytes[11], however, the importance of Nr4a1 in the mucosal lamina propria of the intestine remains unknown. The monocyte switching to macrophages may be impaired in models of IBD with monocytes retaining their inflammatory phenotype leading to intestinal damage[8]. This has led to the general conclusion that monocytes are pathogenic in sterile injury. However, in some models of inflammation, $Ccr2^{-/-}$ mice lacking circulating monocytes, had impaired resolution suggesting a beneficial role for these cells[12] and begging the question what role the monocyte/macrophage lineage plays in homeostasis and inflammatory processes.

An added complexity within the intestine is the fact that there are different layers of the intestinal tract and as such anatomical localization within the intestine likely reflects functional differentiation of macrophages in each layer. Indeed, the macrophages within the muscularis region closely apposed to the enteric nervous system support the neurons and contribute to both intestinal secretion and motility[13], while the macrophages in the submucosa maintain the integrity of the vasculature in this layer[14]. These two populations of macrophages are at a significant distance from the lumen of the bowel, may not all be replaced from bone marrow[14,15] and as such may not perform the same functions as the very large population of mucosal lamina propria macrophages that reside adjacent to the billions of commensal bacteria with which there may be direct cross-talk[6,16].

The CX3CR1$^+$ macrophages in the mucosal lamina propria increase in number from small intestine to colon[17,18] and have been shown to have phagocytic and bactericidal activity[8,19]. It seems reasonable that they would play a key barrier function to potential invaders. However, not all evidence supports this as Schreiber et al., revealed that mice that lacked DCs but not CX3CR1$^+$ macrophages succumbed to $Citrobacter$ $rodentium$ infection within 10 days[20]. In fact while DCs sample antigens from the intestinal vasculature to present antigens to lymphocytes, patrol the epithelial barrier by crossing back and forth across the epithelium and migrate back to lymph nodes to present antigens, macrophages perform none of these functions[21–25] raising questions about their specific functions.

## Results

**CCR2$^+$ monocytes form a perivascular network of CX3CR1$^+$ macrophages.** Under basal conditions, flow cytometry analysis revealed that the majority of CX3CR1$^+$ cells in colonic mucosal lamina propria were F4/80$^+$CD11b$^+$ macrophages (Supplementary Fig. 1a, b) as previously described[17,18]. Intravital imaging revealed a dense network of CX3CR1$^+$ macrophages in villi of the small intestine (Fig. 1a) and a complex honeycomb lattice structure in the rest of the mucosal lamina propria of the small bowel and the large bowel (Fig. 1a). To understand this lattice network, we used higher magnification imaging with 3-D reconstruction which revealed that this population of large elongated CX3CR1$^+$ (CCR2$^{lo}$ or negative) macrophages were sessile and avidly surrounded the vasculature (Fig. 1b, c). We identified CX3CR1$^{intermediate (int)}$ and CX3CR1$^{hi}$ macrophages using flow cytometry and 20% of both populations showed CD80$^-$CD206$^+$ phenotype (Supplementary Fig. 1c, d). It is worth mentioning that the CX3CR1$^{int}$ macrophages are likely immature and make up less than one-third of the total population of CX3CR1$^+$ macrophages. This is consistent with continuous turnover of these cells.

Flow cytometry revealed a spectrum or waterfall of monocytes that ranged from CCR2$^{hi}$CX3CR1$^+$ to CX3CR1$^+$CCR2$^{lo}$ (Supplementary Fig. 1e) as previously described in refs. [8,10]. Use of $Cx3cr1^{GFP/+}Ccr2^{RFP/+}$ reporter mice revealed that only classical CCR2$^{hi}$CX3CR1$^+$ monocytes and not non-classical CX3CR1$^+$CCR2$^{lo}$ monocytes were seen adhering in the intestinal vasculature. Many of these round-shaped CCR2$^+$ monocytes took up residency outside the vasculature, directly adjacent and in contact with CX3CR1$^+$ macrophages (Fig. 1d, e). Red monocytes can be seen juxtaposed to the sessile green macrophages (Supplementary Video 1). Further characterization of these tissue CCR2$^+$ monocytes using flow cytometry revealed loss of Ly6C and imaging revealed an orange and yellow appearance (subtle increase in CX3CR1 and less and less CCR2) consistent with a changing phenotype in these monocytes in the colonic mucosa. Similar observations were noted in the small intestine (Fig. 1f, g). From the flow cytometry data, it appears that it is more a loss of CCR2 and associated RFP signal than a change in CX3CR1 and a GFP signal that constitutes the red, orange, yellow and green colors. For completeness, it is worth mentioning that there also exists a population of CD11b$^-$CX3CR1$^-$CCR2$^+$ population of CD45$^+$ cells that were made up primarily of CD3$^+$ T cells as well as a few DCs, basophils, and NK cells (Supplementary Fig. 1f).

Bone marrow (BM) cells were harvested from $Cx3cr1^{GFP/+}Ccr2^{RFP/+}$ mice and were transplanted into C57BL/6 mice after sublethal whole body irradiation. The irradiation did not affect the resident population of CX3CR1$^+$ macrophages in the mucosal lamina propria as assessed in $Cx3cr1^{GFP/+}Ccr2^{RFP/+}$ mice (Supplementary Fig. 2a) while the monocyte numbers were reduced by close to 50% (Supplementary Fig. 2b). Accumulation of new BM-derived CCR2$^{hi}$CX3CR1$^+$ monocytes was observed in colonic mucosal lamina propria at 2 weeks after bone marrow transplantation (BMT) whereas very few CX3CR1$^+$CCR2$^{lo}$ monocytes/macrophages were detected at this time (Fig. 2a–c). These reporter monocytes in colonic mucosal lamina propria changed from red to orange to yellow to green over time, signifying a change of phenotype from CCR2$^{hi}$CX3CR1$^+$ to CX3CR1$^+$CCR2$^{lo}$ monocytes (Fig. 2a, b). Complete turnover to fluorescent BM derived macrophages took 6–8 weeks. The data support conversion of cells in lamina propria rather than recruitment of different types of monocytes at different times, as CCR2$^{hi}$CX3CR1$^+$ monocytes but not CX3CR1$^+$CCR2$^{lo}$ monocytes adhered in the vasculature and emigrated into the lamina propria. Also, in rare cases, we could image in real-time changes of monocytes from round yellow to more elongated

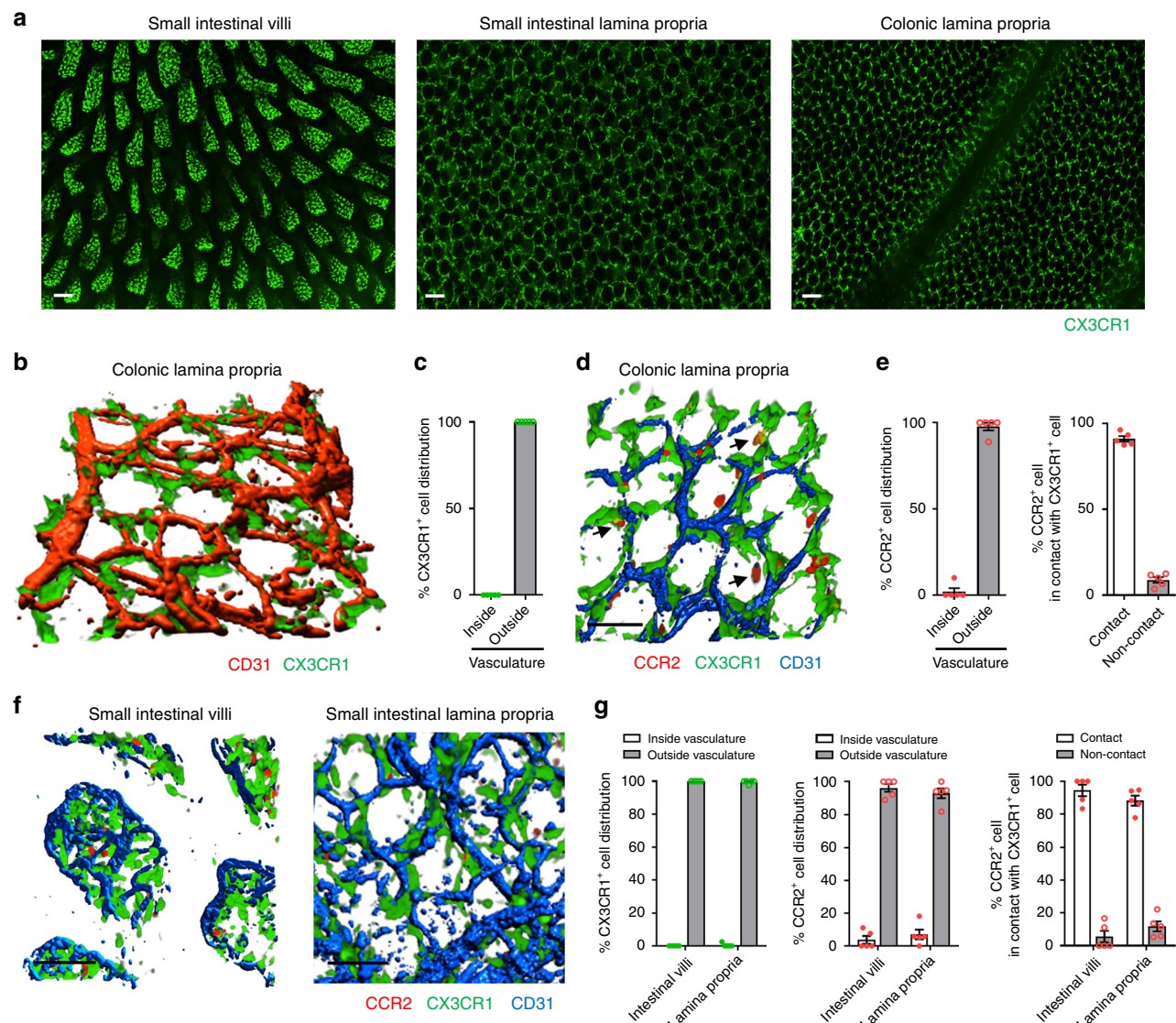

**Fig. 1 Intestinal CX3CR1⁺ macrophages and CCR2⁺ monocytes constitute the close network that surrounds vasculature. a** Representative stitched images (from 12 different fields of view) of small intestinal villi (left), small intestinal lamina propria (LP) (middle), and colonic LP (right) in $Cx3cr1^{GFP/+}$ mice. Scale bars, 100 μm. **b** Representative three-dimensional (3D) image of CX3CR1⁺ macrophages (green) in colonic LP and **c** quantification of their distribution. $n = 5$. **d** Representative 3D image of colonic LP CX3CR1⁺ macrophages and CCR2⁺ cells (red) at basal condition in $Cx3cr1^{GFP/+}Ccr2^{RFP/+}$ mouse. Arrows indicate orange/yellow cells. Scale bar, 50 μm. (**e**) Localization of colonic LP CCR2⁺ cells and their positional relationships with CX3CR1⁺ cells. $n = 5$. (f) Representative 3D images of small intestinal villi (left) and LP (right). Scale bars, 50 μm. **g** Localization of small intestinal CX3CR1⁺ and CCR2⁺ cells in villi and LP and their positional relationships. $n = 5$. Data represent mean ± SEM. Source data are provided as a Source Data file.

CX3CR1⁺ (green) macrophages that would localize around blood vessels over about 45–60 min (Fig. 2d). This was very difficult to see by eye but hue analysis confirmed rapid transition (Fig. 2e). This suggests that complete turnover of macrophages takes up to 6–8 weeks as previously reported[14,15], but individual monocytes can convert rapidly to macrophages as needed. $Ccr2^{-/-}$ mice have a dearth of classical monocytes in the circulation due to difficulty egressing from BM[26]. We used $Cx3cr1^{GFP/+}Ccr2^{RFP/RFP}$ mice to demonstrate that following 6 weeks of BMT these CCR2-deficient mice had a significant reduction in elongated green macrophages (Fig. 2f, g). While there were fewer orange and yellow monocytes (Fig. 2g), this did not achieve significance due to the small numbers of these cells in all mice at any time examined. Although some investigators have reported a role for the CX3CR1 ligand in the recruitment of progenitor macrophages in some tissues, imaging of CX3CR1-deficient mice showed equivalent macrophage distribution to wild-type mice, suggesting

the number and localized interdigitation of macrophages is independent of CX3CR1 ligands (Supplementary Fig. 3a–d).

Since Nr4a1 was described as a key transcription factor that permitted differentiation of CCR2^hiCX3CR1⁺ monocytes to CX3CR1⁺CCR2^lo monocytes[11], we used $Nr4a1^{-/-}Cx3cr1^{GFP/+}Ccr2^{RFP/+}$ mice and performed BMT. Imaging these mice at 6 weeks after BMT showed a delay in macrophage turnover in colonic mucosal lamina propria, despite normal recruitment of the CCR2^hi (red) monocytes (Fig. 2f, g). We also confirmed the delayed appearance of both immature CX3CR1^int and mature CX3CR1^hi macrophages by flow cytometry (Fig. 2h). Interestingly, while the numbers of CX3CR1 macrophages were marginally reduced (Fig. 2h) in $Nr4a1^{-/-}$ mice, the imaging revealed that the macrophages less avidly surrounded the vasculature forming a less contiguous lattice network at 6 weeks with gaps between individual macrophages (Fig. 2f), suggesting that Nr4a1 helps complete the full anatomical maturation of these cells.

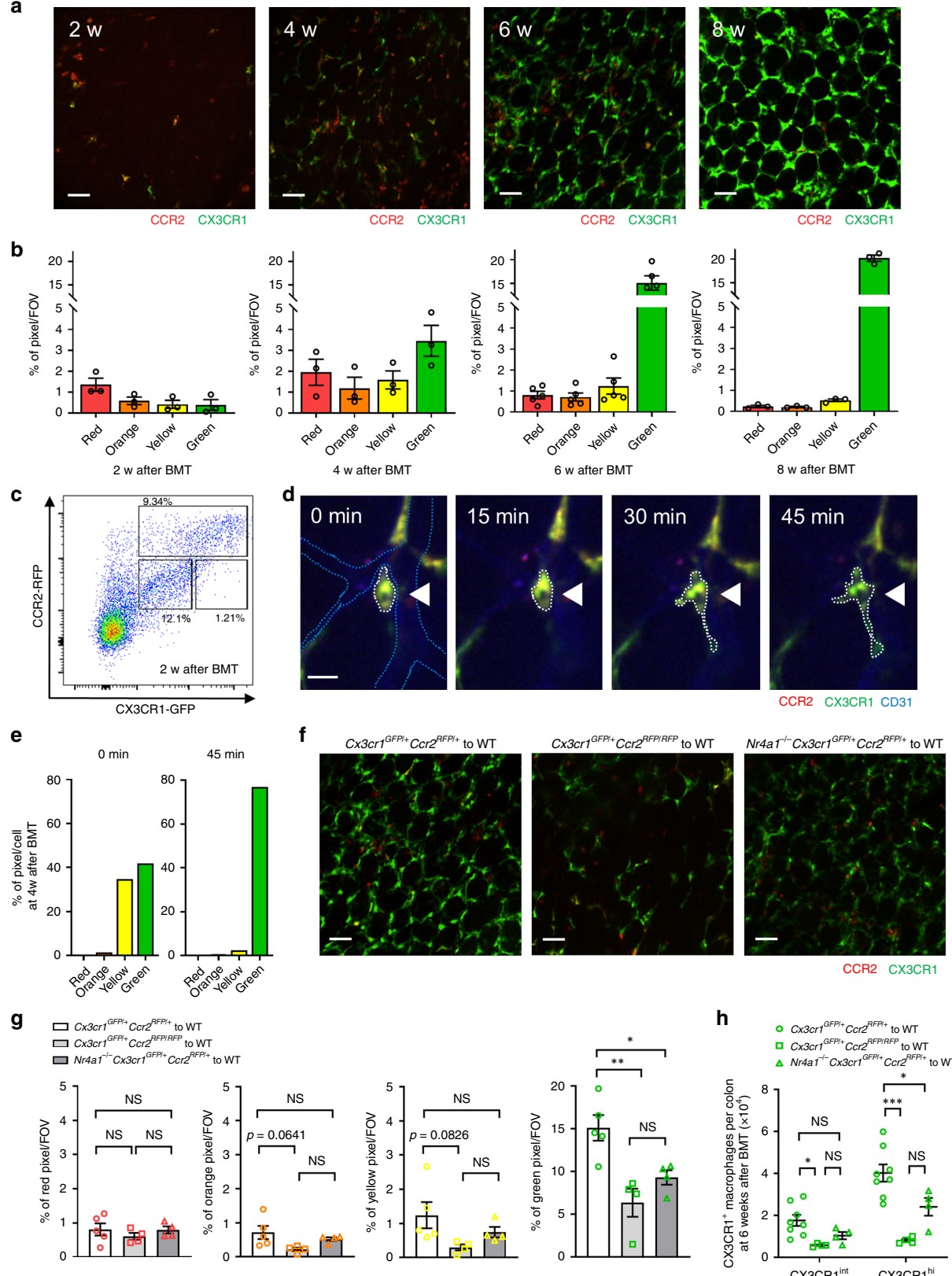

**Microbiome is key to perivascular macrophage localization.** To investigate the role of the microbiome, mice were given a cocktail of broad-spectrum antibiotics[27] from before birth until 8 weeks of age which dramatically reduced the intestinal bacterial load by more than 99.6% (Supplementary Fig. 4a, b). The CX3CR1+ macrophages covered less surface area in the villi of the small intestine and the lamina propria of the colon in Abx-treated mice (Fig. 3a–c and Supplementary Fig. 4c–f). Abx treatment did not affect the development of intestinal vasculature (Fig. 3d). In the colon it was very evident that the macrophages were "holding

**Fig. 2 Turnover of intestinal macrophage is dependent on monocyte conversion. a** Representative images of colonic LP in C57BL/6 mice 2, 4, 6, and 8 weeks after BMT from $Cx3cr1^{GFP/+}Ccr2^{RFP/+}$ mice. Scale bars, 50 μm. **b** Analysis of monocyte hues at the indicated time points. $n = 3-5$ in each time points. **c** Flow cytometry analysis of colonic LP CCR2$^+$ and CX3CR1$^+$ cells 2 weeks after BMT. Cells were pregated on size, viability, and CD45$^+$. Data are representative of three independent experiments. **d** Time-lapse images demonstrate the change in color and shape of CX3CR1$^+$CCR2$^+$ cell 4 weeks after BMT (white arrow heads; round yellow to elongated green). White dotted lines exhibit the outline of indicated cell. Blue dotted lines indicate CD31$^+$ vasculature. Scale bar, 20 μm. **e** Analysis of monocyte hues of the indicated cell in **d** at 0 and 45 min. **f** Representative images of colonic LP of C57BL/6 mice 6 weeks after BMT from $Cx3cr1^{GFP/+}Ccr2^{RFP/+}$, $Cx3cr1^{GFP/+}Ccr2^{RFP/RFP}$, or $Nr4a1^{-/-}Cx3cr1^{GFP/+}Ccr2^{RFP/+}$ mice. Scale bars, 50 μm. **g** Quantification of monocyte hues 6 weeks after BMT in each group. $n = 3-5$ per group. **h** Quantification of the number of CX3CR1$^{int}$ and CX3CR1$^{hi}$ macrophages by flow cytometry at 6 weeks after BMT in each group. Cells were gated on size, viability, CD45$^+$, CD103$^-$, CD11b$^+$, F4/80$^+$ and CX3CR1$^{int/hi}$. $n = 4-8$ per group. Data represent mean ± SEM. $*p < 0.05$, $**p < 0.01$, $***p < 0.001$, NS not significant. Source data are provided as a Source Data file.

hands" interdigitated around the vasculature of the LP in untreated mice whereas in mice treated with antibiotics there were clear gaps between the CX3CR1$^+$ macrophages (Fig. 3e). Higher magnification combined with 3D reconstruction revealed that macrophages were in contact and interdigitated with each other in untreated mice forming a barrier around the vasculature such that the vasculature was very difficult to see en face from the luminal side of the colon in untreated mice but not in Abx-treated animals (Fig. 3f).

Interestingly, the CX3CR1$^+$ macrophages did not form an interdigitated anatomical barrier in the submucosa and muscularis layers in wild-type mice (Fig. 3b, c and Supplementary Fig. 4c–f). In the submucosa, the macrophage sparsely abutted the vasculature whereas in the muscularis the macrophages were strikingly different having dendrites and did not appear anywhere near the vasculature. Macrophages of Abx-treated and untreated mice had an identical distribution pattern in the submucosa and muscularis.

To ensure that the antibiotics were not having a microbiome-independent influence, we also generated germ-free CX3CR1-GFP mice and observed very similar paucity of macrophages in the mucosal layer throughout the intestinal tract, revealing clear gaps in the macrophage barrier (Fig. 3g). Although germ-free mice have been reported to have less dense vasculature in the mid-distal part of the small intestine[28], the colon of germ-free and SPF mice, had identical intestinal vasculature density so that alterations in vasculature could not account for the macrophage distribution difference (Fig. 3g). What was striking was the clear adhesion of the macrophages to the lamina propria microcirculation, but the spectacular retraction of the perivascular macrophages exposing large areas of the vasculature in the absence of a microbiota (Fig. 3f, g). Flow cytometry analysis confirmed that there were less colonic CX3CR1$^{int}$ and CX3CR1$^{hi}$ macrophages in Abx-treated mice (Fig. 3h, i) as previously described using germ-free mice[6] but did not affect the CD80$^+$CD206$^-$ or CD80$^-$CD206$^+$ phenotype of CX3CR1$^+$ macrophages at steady state (Fig. 3i). Importantly, Abx-treated mice showed no reduction in monocytes in blood suggesting that the decrease in macrophages in intestine was also not due to the blood compartment monocytes (Supplementary Fig. 5a, b).

To examine the impact of the microbiota on turnover of macrophages, we imaged the colonic mucosal lamina propria of Abx-treated C57BL/6 mice at 2, 4, 6, and 8 weeks after BMT from $Cx3cr1^{GFP/+}Ccr2^{RFP/+}$ mice (Fig. 4a). Imaging showed a delay in CCR2$^+$ monocyte accumulation in Abx-treated mice at 2 weeks following BMT, followed by a significant delay in the appearance of CX3CR1$^+$ monocytes/macrophages over the next 6 weeks (Fig. 4a, b). Flow cytometry data of colonic mucosal lamina propria cells 2 and 6 weeks after BMT confirmed these results (Fig. 4c, d). This was not due to a defect in BM or blood of Abx-treated mice as there were no significant differences in the total number of leukocytes in these compartments (Supplementary Fig. 5c, d). Interestingly, the delayed turnover of macrophages in Abx-treated mice was partially restored by oral administration of

high quantities of pathogenic heat-killed bacteria including *Staphylococcus aureus*, *Escherichia coli*, and *Salmonella typhimurium* given weekly for 6 weeks (Supplementary Fig. 6a, b) but not by simply giving back individual TLR agonists LTA, LPS, or CpG after BMT (Supplementary Fig. 6c, d). Additionally, turnover of macrophages was also not inhibited when BMT was performed under SPF conditions using $MyD88^{-/-}$ or $MyD88^{-/-}Trif^{-/-}$ mice as a donor or a recipient (Supplementary Fig. 6e–g).

Intriguingly, when Abx water was administered to mice receiving BMT from $Cx3cr1^{GFP/+}Ccr2^{RFP/RFP}$ mice lacking 90% of classical monocytes in the blood compartment, mucosal lamina propria macrophage turnover was completely inhibited at 6 weeks (Fig. 4e, f) demonstrating that CCR2 ligands and chemoattractants either induced by the microbiome or metabolites from the microbiome together accounted for all recruitment of monocytes to the gastrointestinal tract at the early time points. At six weeks there was finally arrival of monocytes in $Ccr2^{-/-}$ mice treated with antibiotics but still no orange, yellow or green cells suggesting a huge delay in monocyte recruitment/conversion. Absence of $Nr4a1$, did not further delay turnover of macrophages beyond the profound delay with Abx-treated mice receiving BMT (Fig. 4e, f). This suggests that the microbiome helps to convert classical monocytes to intestinal macrophages by activating Nr4a1.

**The perivascular macrophages are a barrier to bacterial pathogens.** Figure 5a, b showed very effective coverage of the mucosal vasculature of the small intestine and colon in SPF mice. Following Abx treatment for 8 weeks, there were clear gaps between individual perivascular macrophages (Fig. 5a, b) suggesting breakdown of a potential barrier. We used both *S. aureus* and *S. typhimurium* as model organisms. It is worth mentioning that while *S. aureus* is not a major intestinal pathogen in adults, it is directly relevant to pediatric populations which are frequently colonized by *S. aureus* following antibiotics usage and can cause severe disease in these young patients[29]. We gavaged mice with a fluorescently tagged live *S. aureus* 2 days after terminating antibiotics and looked for bacteria in intestinal draining lymph nodes as well as the liver and spleen, the two organs that filter bacteria from blood. While similar amounts of *S. aureus* were seen in the intestinal tract of Abx-treated and untreated mice, there was significant dissemination of this fluorescent pathogen in the liver and spleen but only in the Abx-treated mice (Fig. 5c). By contrast, the intestinal vascular barrier of the SPF mice that received no Abx appeared to be impenetrable as no bacteria were seen in liver or spleen. The *S. aureus* could only be detected in the blood filtering organs (liver and spleen) at 24 h but not at the early 6 h timepoint (Fig. 5c) in antibiotic-treated mice. Imaging the liver revealed significant fluorescent bacteria in Abx-treated mice (Fig. 5d). In numerous cases we found bacteria in the liver and spleen, but not the mesenteric lymph nodes suggesting the bacteria were entering directly into the circulation rather than leaving via the lymphatics. We detected the perivascular CX3CR1$^+$ macrophages catching *S. aureus* that had invaded across the

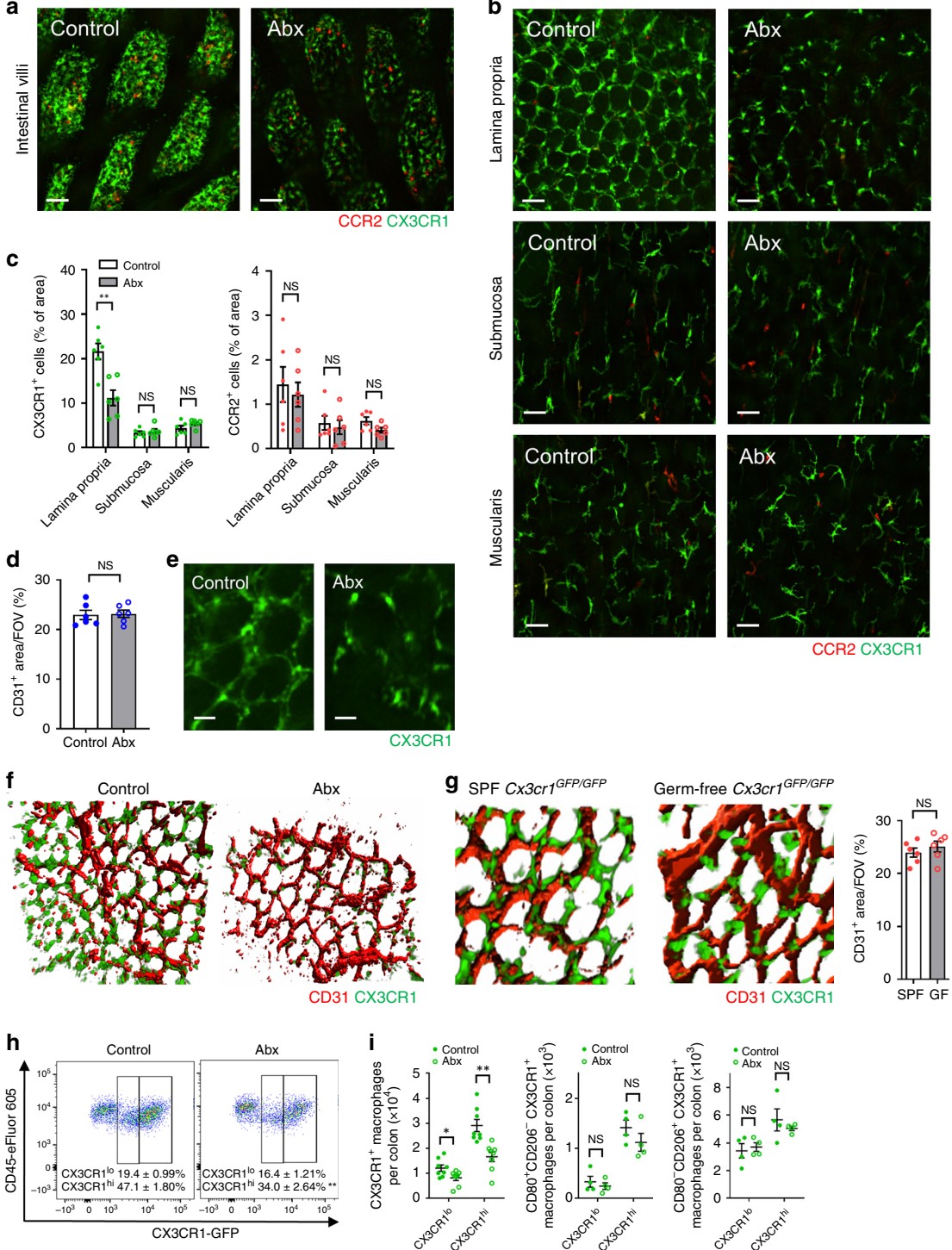

**Fig. 3 Gut microbiota affects the distribution and morphology of intestinal CX3CR1⁺ macrophages.** Representative images of **a** small intestinal villi and **b** colonic lamina propria (LP), submucosa and muscularis in *Cx3cr1*^GFP/+^*Ccr2*^RFP/+^ mice (The left column is SPF control, the right column is Abx-treated mice). Scale bars, 50 µm. **c** Quantification of CX3CR1⁺ cells or CCR2⁺ cells per field of view (FOV) in each layer of the colon in control and Abx-treated mice. *n* = 6 per group. **d** Quantification of CD31⁺ area per FOV in colonic LP in control and Abx-treated mice. *n* = 6 per group. **e** Representative high-magnification and **f** 3D reconstructed images of CX3CR1⁺ macrophages and vasculature in LP. Scale bars, 20 µm. **g** Representative 3D reconstructed images of CX3CR1⁺ macrophages in LP in control and germ-free *Cx3cr1*^GFP/GFP^ mice. Quantification of CD31⁺ area per FOV in colonic LP in control and germ-free mice (right). *n* = 6 per group. **h** Flow cytometry analysis of the proportion of CX3CR1^int^ and CX3CR1^hi^ macrophages in the colonic LP in control and Abx-treated mice at steady state. Cells were pregated on size, viability, CD45⁺, CD103⁻, CD11b⁺, and F4/80⁺. **i** Absolute number of CX3CR1⁺, CD80⁺CD206⁻, and CD80⁻CD206⁺ macrophages in the total colon of control and Abx-treated mice. Cells were pregated on size, viability, CD45⁺, CD103⁻, CD11b⁺, and F4/80⁺. *n* = 4–8 per group. Data represent mean ± SEM. *p < 0.05, **p < 0.01, NS, not significant. Source data are provided as a Source Data file.

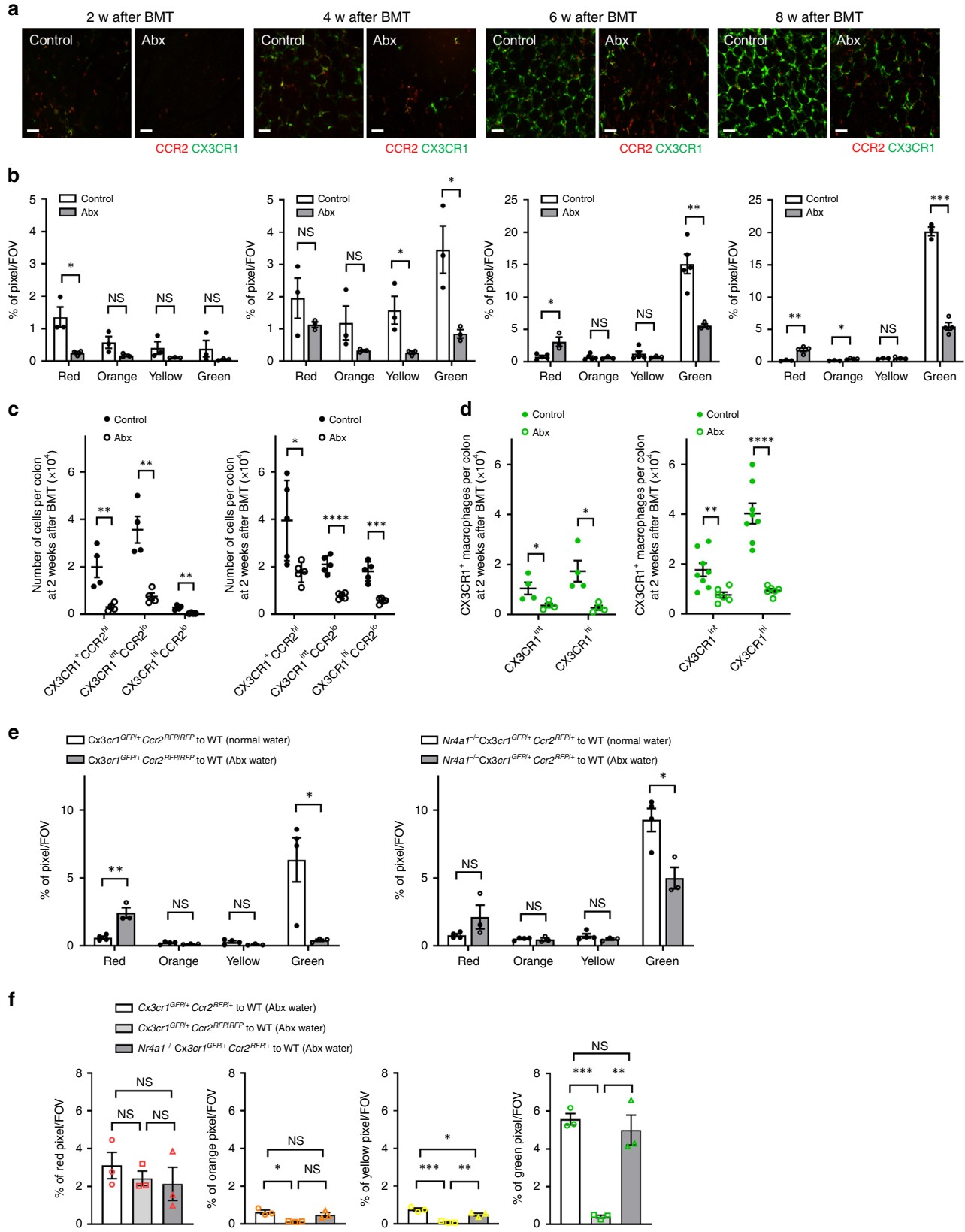

colonic epithelium (Fig. 5e), but this was a rare event. However, when we overtly disrupted the epithelial barrier using DSS and allowed more bacteria into the lamina propria, we observed many CX3CR1+ macrophages had captured significant numbers of mCherry-*S. aureus* (Supplementary Fig. 7a–d). The massive

injury with DSS allowed *S. aureus* to bypass the macrophages and be detected as early as 6 h in the liver, spleen and kidney (Supplementary Fig. 7e).

We also examined a more common adult intestinal pathogen, namely *Salmonella typhimurium* which has much greater capacity

**Fig. 4 Depletion of gut microbiota impairs the turnover of intestinal macrophages. a** Representative intravital images of colonic LP in SPF control and Abx-treated mice at 2, 4, 6, and 8 weeks after bone marrow transplantation (BMT) from $Cx3cr1^{GFP/+}Ccr2^{RFP/+}$ mice. Scale bars, 50 μm. **b** Analysis of monocyte hues in control and Abx-treated mice at the indicated time points. $n = 3$–5 in each time points. **c** The number of CX3CR1+CCR2hi cells, CX3CR1intCCR2lo cells, and CX3CR1hiCCR2lo cells per colon at 2 and 6 weeks after BMT quantified by flow cytometry. Cells were pregated on size, viability, and CD45+. $n = 4$–5 per group. **d** Quantification of the number of CX3CR1int and CX3CR1hi macrophages at 2 and 6 weeks after BMT. Cells were pregated on size, viability, CD45+, CD103−, CD11b+, and F4/80+. $n = 4$–8 per group. **e, f** Quantification of monocyte hues of control and Abx-treated mice at 6 weeks after BMT from $Cx3cr1^{GFP/+}Ccr2^{RFP/+}$, $Cx3cr1^{GFP/+}Ccr2^{RFP/RFP}$ or $Nr4a1^{-/-}Cx3cr1^{GFP/+}Ccr2^{RFP/+}$ mice. $n = 3$–5 per group. Data represent mean ± SEM. *$p < 0.05$, **$p < 0.01$, ***$p < 0.001$, ****$p < 0.0001$, NS not significant. Source data are provided as a Source Data file.

to cross the epithelial barrier. We isolated a loop of colon and when mCherry-*S. typhimurium* was injected into this isolated loop it readily crossed the epithelial barrier similarly in both untreated and Abx-treated mice (Fig. 5f, g). *S. typhimurium* was taken up avidly by CX3CR1+ macrophages surrounding the vasculature in untreated mice (where green macrophages have red bacteria giving a yellow punctate appearance) (Fig. 5h left panel). When transparency was increased (removing the green macrophage signal), many red bacteria come into view highlighting the many *S. typhimurium* taken up by these cells in untreated animals (Fig. 5h right panel). In Abx-treated animals, red *S. typhimurium* could be seen adhering to the blue vasculature in between the CX3CR1+ GFP-positive macrophages (Fig. 5i). In Fig. 5i, one can see bacteria abutting the vasculature and over 30 s suddenly disappearing into the mainstream of blood. Supplementary Video 2 clearly shows the large gaps between CX3CR1+ GFP-positive macrophages in Abx-treated mice through which red *S. typhimurium* can be seen accessing the vasculature.

To determine the specific role of macrophages in *S. typhimurium* translocation, we generated $Csf1r^{LsL-DTR}/LysM^{cre}$ mice which permit selective depletion of macrophages but not dendritic cells or other cell types in the intestines[20]. Administration of diphtheria toxin (DT) depleted over 70% CX3CR1+ macrophages in the colon (Fig. 5j). In this setting, more dissemination of *S. typhimurium* to the liver and spleen was observed while there were no differences in the MLN (Fig. 5k), consistent with a direct translocation into blood rather than via the draining lymphatics. These mice also had a significant depletion of macrophages in liver and spleen and yet there were more bacteria in these organs underscoring the number of bacteria that translocated into blood when the intestinal perivascular macrophages were depleted.

**Nr4a1 and the microbiome mature intestinal macrophages for repair.** In addition to bacterial capture, we also examined the role of macrophages in thermal ablation of all cells in a 500 μm focal injury. This model allowed us to image recruitment and conversion of monocytes to macrophages in an area eradicated of resident cells. Imaging and flow cytometry revealed that CD11b+Ly6C+CX3CR1+CCR2hi monocytes start infiltrating into the injury site within 6 h (Fig. 6a–e). CX3CR1+ macrophages that were adjacent to the injury site but localized in healthy tissue remained sessile and did not leave their post to directly invade the site of injury. Instead, accumulated CCR2hiCX3CR1+ monocytes began to transition into CX3CR1+CCR2lo monocytes/macrophages within 24 h and were entirely CX3CR1+CCR2lo within the injury site within the first 96 h (Fig. 6a, b). The majority of the cells were still yellow, highlighting remnant CCR2-RFP within these CX3CR1-GFP cells. Flow cytometry of biopsies of injured intestinal tissue as early as 6 h post injury revealed a significant increase in the number of CD11b+Ly6C+CCR2hi monocytes but not CX3CR1+CCR2lo monocytes or other CCR2+ cells (Fig. 6d), highlighting a much bigger ratio of Ly6Chi to CX3CR1+ cells at this early time point (Fig. 6e). Imaging 7 days after injury showed that blood vessels had regrown and the injury site was completely

repopulated with CX3CR1+ cells and 80% were contacting the newly reconstructed blood vessels (Fig. 6f, g).

To determine a role for CCR2, Nr4a1 and the microbiome in emergency repopulation we imaged $Cx3cr1^{GFP/+}Ccr2^{RFP/RFP}$, $Nr4a1^{-/-}Cx3cr1^{GFP/+}Ccr2^{RFP/+}$, and antibiotic-treated mice under basal conditions and during tissue injury. Imaging and flow cytometry revealed that $Cx3cr1^{GFP/+}Ccr2^{RFP/RFP}$ mice, but not $Nr4a1^{-/-}Cx3cr1^{GFP/+}Ccr2^{RFP/+}$ mice showed a decrease in the number of colonic mucosal lamina propria CX3CR1hi macrophages under basal conditions (Supplementary Fig. 8a–c). Following the thermal injury in wild type, $Ccr2^{-/-}$, and $Nr4a1^{-/-}$ mice, emergency repopulation of CCR2hi monocytes was severely impaired in $Ccr2^{-/-}$ mice. This translated into a delay in appearance of CX3CR1+ monocytes and macrophages at all time points examined (Supplementary Fig. 8d, e). By contrast, $Nr4a1^{-/-}Cx3cr1^{GFP/+}Ccr2^{RFP/+}$ mice did not show an inhibition of CCR2hi monocyte recruitment at 6 or 24 h post injury, but the subsequent monocyte conversion at 24 h to orange monocytes and at 48 h to yellow monocytes was significantly reduced suggesting a delay in conversion within the injury site (Supplementary Fig. 8d, e). The recruitment and conversion of monocytes to CX3CR1+CCR2lo monocytes/macrophages were critical for tissue repair in a traumatic sterile injury including removal of necrotic cells and revascularization of injured tissue (Supplementary Fig. 8f, g). Vascular permeability impairment was less severe in the $Nr4a1^{-/-}$ mice than the $Ccr2^{-/-}$ mice, suggesting that even the classical monocytes that are recruited in $Nr4a1^{-/-}$ mice have some contribution to healing.

We next evaluated the emergency repopulation of macrophages and the repair process in Abx-treated mice. Interestingly, intravital imaging revealed that the early 6 h monocyte recruitment and the progression of conversion towards CX3CR1+ monocytes/macrophages were consistently delayed by about 24 h at each time point in Abx-treated mice until 96 h. Macrophage numbers were identical by 168 h (Fig. 7a, b). Flow cytometry analysis also revealed some delay in the proportion of mature CX3CR1+ macrophages per CD45+CD103− cells in injured colon in Abx-treated mice (Fig. 7c, d). Intriguingly, the absence of the microbiome blocked the CD80−CD206+ phenotype in CX3CR1hi macrophages within the injury site (Fig. 7d). These markers represent a more repair like macrophage, and their absence in Abx-treated mice associated with significantly impaired necrotic cell clearance and revascularization within the injury site (Fig. 7e–h). Moreover, Abx-treated mice showed higher vascular permeability 4 days post injury, suggesting an immature and/or dysfunctional vasculature (Fig. 7i, j). Supplementary Video 3 highlights the very significant and rapid transendothelial movement of flourescent TRITC in Abx-treated mice.

**Dysbiotic mice not $Nr4a1^{-/-}$ mice have delayed intestinal repair in DSS colitis.** Previous studies have shown that germ-free and Abx-treated mice are more susceptible to DSS-induced colitis[30,31]. Our data also revealed an increased impairment when DSS treated mice received Abx, but the underlying mechanism

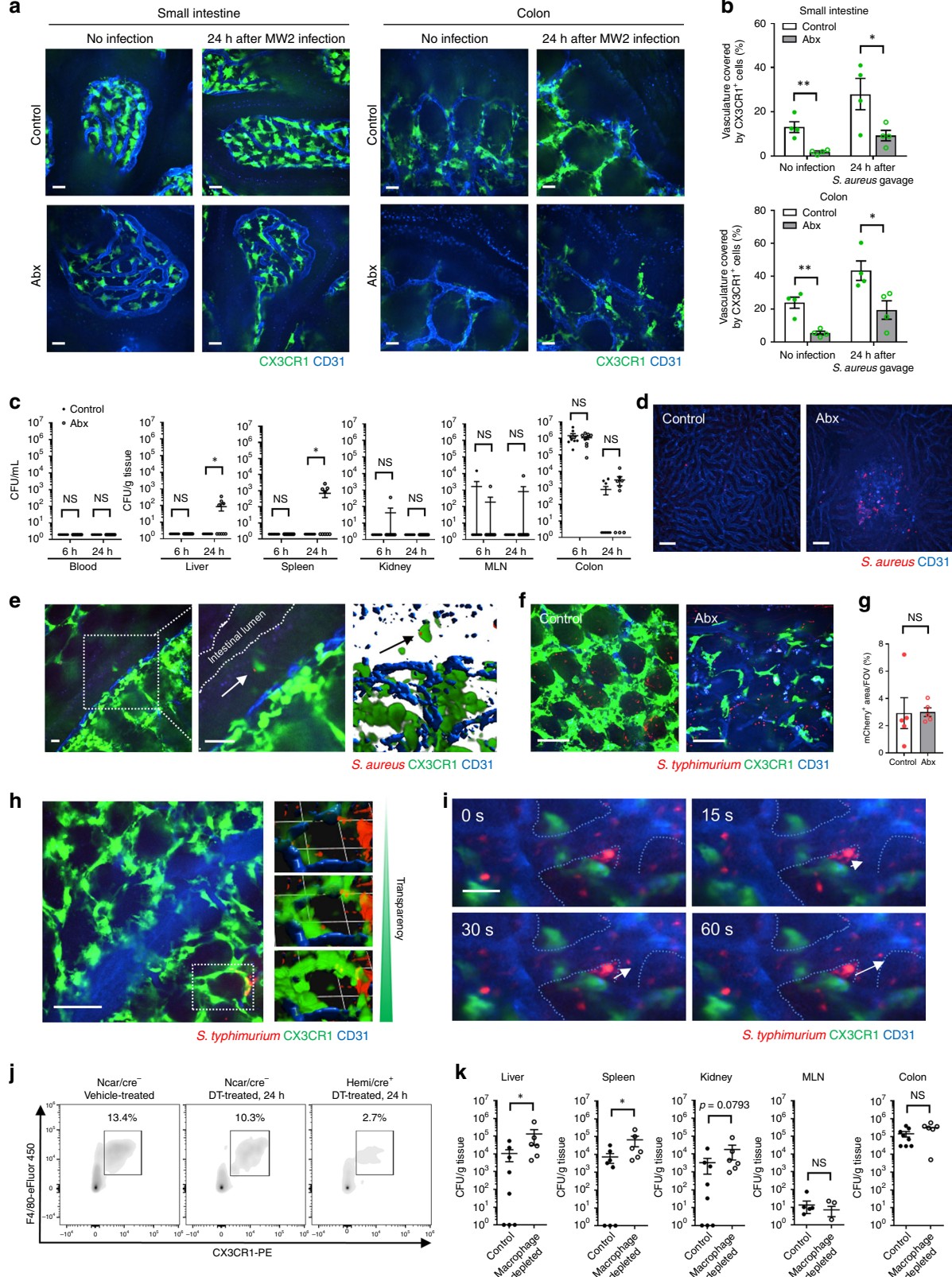

was not increased inflammation. Indeed, the shortening of the colon (Fig. 8a), used as a marker of inflammatory injury as well as all of the pro-inflammatory cytokines and chemokines were lower in the Abx-treated mice that received DSS than in DSS treated mice that received no Abx (Fig. 8b, c) confirming previous results[31]. Moreover, Abx-treated mice had reduced induction of

MCP-1 (CCL2), the key chemokine that attracts CCR2[+] monocytes, in the colon of DSS treated mice (Fig. 8c). In addition, TNF-α and IL-6, cytokines known to be mainly produced by macrophages and promote regeneration of damaged intestinal mucosa by acting on epithelial cells[32] were reduced by 90% (Fig. 8b). When we imaged the lamina propria of control and

**Fig. 5 CX3CR1+ macrophages in intestinal lamina propria of antibiotic-treated mice do not cover the vasculature and result in bacterial dissemination.**
**a** Representative images of small intestinal villi and colonic lamina propria in control and Abx-treated $Cx3cr1^{GFP/+}$ mice at 24 h after the gavage of *S. aureus* MW2. Scale bars, 20 μm. **b** Quantification of area of vasculature covered by CX3CR1+ macrophages in control and Abx-treated $Cx3cr1^{GFP/+}$ mice with or without *S. aureus* infection. **c** CFU counts at 6 and 24 h after *S. aureus* MW2 gavage in control and Abx-treated mice. $n = 9$–10 per group. **d** Still image of the liver in control and Abx-treated mice at 24 h after gavage of mCherry-MW2. Scale bars, 50 μm. **e** Representative images of CX3CR1+ macrophage (green) catching *S. aureus* (red) in colonic epithelium at 24 h after the gavage of *S. aureus* mCherry-MW2. Higher magnification of the indicated area was shown in middle and 3D image was shown in right. Arrows indicate *S. aureus*. Scale bars, 20 μm. **f** Representative images of colonic lamina propria in control and Abx-treated $Cx3cr1^{GFP/+}$ mice at 6 hrs after the intraluminal injection of mCherry-*Salmonella typhimurium*. Scale bars, 50 μm. **g** Quantification of mCherry+ area in control and Abx-treated $Cx3cr1^{GFP/+}$ mice. **h** Images (right; 3D reconstruction) of mCherry-*S. typhimurium* caught by intestinal CX3CR1+ macrophages in SPF control mice. Transparency adjustment was applied (upper) to allow for visualization of *S. typhimurium* inside CX3CR1+ macrophages. Scale bar, 50 μm. 1 Unit = 23.3 μm. **i** Time-lapse images of colonic lamina propria in Abx-treated $Cx3cr1^{GFP/+}$ mice at 6 h after the intraluminal injection of mCherry-*S. typhimurium*. White arrows indicate *S. typhimurium* disseminating to blood flow. Blue dotted lines indicate CD31+ vasculature. Scale bar, 20 μm. **j** Flow cytometry analysis of the proportion of CX3CR1+ macrophages in the colonic LP in $Csf1r^{LsL-DTR}/LysM^{cre}$ mice with or without diphtheria toxin treatment. Cells were pregated on size, viability, and CD45+. **k** CFU counts at 24 h after *S. typhimurium* ($1 \times 10^9$ CFU) gavage in control and macrophage-depleted mice. $n = 6$−8 per group. Data were pooled from two independent experiments. Data represent mean ± SEM. *$p < 0.05$, **$p < 0.01$, NS not significant. Source data are provided as a Source Data file.

---

Abx-treated $Cx3cr1^{GFP/+}Ccr2^{RFP/+}$ mice 7 days after the start of DSS there was tremendous accumulation of CCR2+ and CX3CR1+ cells in control mice (Fig. 8d). By contrast, Abx-treated mice showed less accumulation of these cells, implying a delay of CX3CR1+ monocytes/macrophages appearance within areas of injury. The reconstruction of blood vessels at the injured mucosal lesion was also inhibited and showed higher permeability, reflecting less mature vasculature in Abx-treated mice (Fig. 8e). Flow cytometry data confirmed these imaging results; both CD80+CD206− and CD80−CD206+ CX3CR1^hi macrophages were decreased in Abx-treated mice (Fig. 8f). Indeed, 7 days after the start of 2% DSS water, all Abx-treated mice demonstrated overt sickness behaviour and all of them died at ~9 days (Fig. 8g). There was very large frank hemorrhage found throughout the intestinal tract when DSS colitis mice received antibiotics (Fig. 8h). This very likely contributed to the reduced survival. This is consistent with the mucosal lamina propria CX3CR1+ macrophages helping in the healing process and in particular in the restitution of a functional vasculature. In the absence of the microbiome, fewer monocytes enter the injury in a timely fashion and repair is delayed.

In a final series of experiments, we also subjected $Nr4a1^{−/−}$ mice to DSS colitis which have previously been reported to be more susceptible to DSS-induced colitis[33]. We found that these mice had delayed conversion of monocytes and fewer recruited macrophages as well as hallmark features of increased injury including delayed recovery of body weight, increased blood in the colon suggesting increased frank hemorrhage but to a much lesser degree than the Abx-treated mice (Fig. 8i–l), such that all $Nr4a1^{−/−}$ mice recovered from DSS colitis. Colon length at 7 days (Fig. 8j) was also not different from wild-type mice. As such, the profound antibiotic-induced exacerbation of injury in DSS colitis was only partially mimicked in mice where the only deficiency was a delay or impairment of monocyte conversion to macrophage.

## Discussion

Macpherson and colleagues have shown that under homeostatic conditions no bacteria were detected in portal blood or in the liver despite bacteria in the draining mesenteric lymph node[34]. The bacteria that had translocated across the intestinal wall were therefore "shunted" via lymphatics to the lymph node and subcapsular sinus macrophages were the primary firewall. The complete lack of bacteria in the portal blood draining the intestines raised the possibility of another barrier preventing bacterial access to the vasculature. Indeed, we have identified another player in this scheme, namely the macrophages in the lamina propria, which construct a primary firewall by forming

interdigitating connections around all of the vasculature of both the small intestine and colon. While intestinal macrophages have been studied extensively, their perivascular localization and their tight connections to form a formidable barrier have not to date been recognized. We show a clear co-localization of the macrophages and vasculature and when we gave *S. aureus* or *S. typhimurium*, we found many of the bacteria inside these perivascular macrophages. Any disruption of the continuous macrophage barrier around the vasculature was associated with increased translocation of bacteria into blood. In some mice, bacteria were detected in the blood but not in lymph nodes suggesting the bacteria entered directly via the blood stream within the intestine, which we observed using intravital microscopy. At no time were these macrophages seen to patrol the mucosal lamina propria or extend their pseudopods into the intestinal lumen. This function may be reserved for the DCs[23]. In addition to barrier function, the perivascular macrophages also appeared to regulate repair especially of a disrupted or ablated vasculature, ensuring limited protein leak from new blood vessels.

It is worth mentioning that these lamina propria macrophages are only one of numerous barriers to bacterial entry from the bowel lumen into the bloodstream. It has been known for a long time that the commensal bacteria, epithelium, IgA, and the mucus in the gut all function as excellent barriers to most bacteria[35]. However, *S. aureus* primarily in newborns[29] and *S. typhimurium* in all humans can bypass the mucosal barrier and enter the parenchyma. Spadoni et al., and Mouries et al., reported that *S. typhimurium* but not *E. coli* could disrupt the endothelial barrier to translocate into the blood stream[21,36] suggesting blood vessels themselves were a microbial barrier. Our work adds a barrier between the epithelium and endothelium, a potent immunological barrier that prevents the bacteria from getting to the endothelium after penetrating the epithelium. It is also possible that the perivascular macrophages, which have direct contact with the blood vessels, could regulate and increase the vascular barrier function. Indeed permeability was greatly increased in new blood vessels during repair when perivascular macrophages were lacking. It is also possible that the perivascular macrophages which have direct contact with the blood vessels could regulate the vascular barrier function increasing its integrity and reducing bacterial translocation in this manner. These macrophages unlike endothelium or other vessel-associated cells, played the unique function of phagocytosing these pathogenic bacteria that crossed the mucosal barrier. Indeed, we found bacteria inside the macrophages surrounding the vessels but not inside endothelium or other vessel-associated cells. Lack of microbiota, resulted in gaps between these perivascular macrophages, and now bacteria could

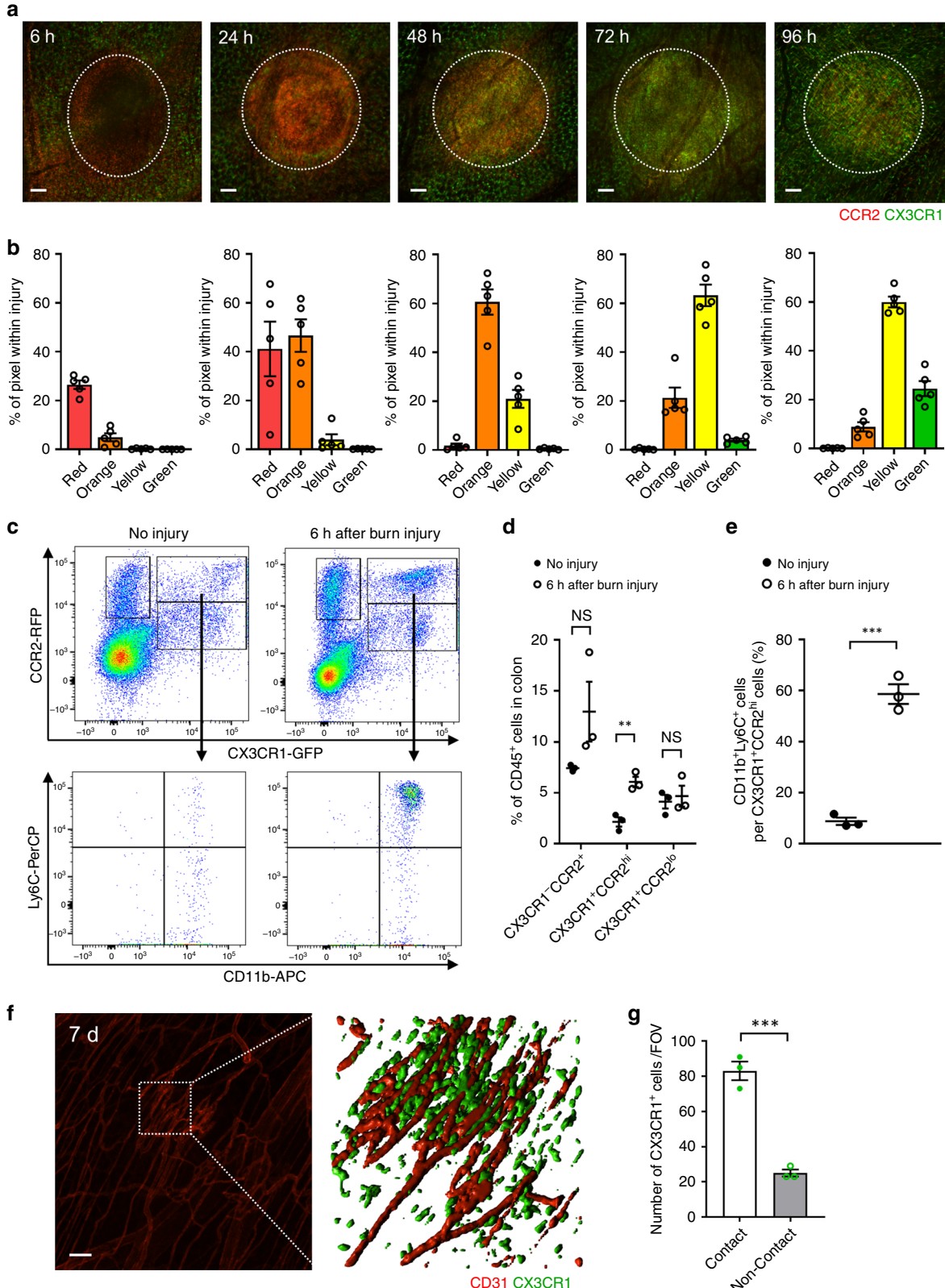

be seen adhering and crossing the vessel wall and entering into the bloodstream. Mouries et al., reported that an altered microbiota due to high fat diet was sufficient to disrupt the endothelial barrier[36] suggesting that both the vascular and macrophage barriers may be affected by microbiota.

Essentially all of our understanding of how the intestine repairs comes from models of inflammatory bowel disease with DSS colitis being the most commonly used system. Based on this work the general view is that classical inflammatory monocytes are recruited into the sites of sterile injury and are impaired in their

**Fig. 6 CD11b$^+$Ly6C$^+$CX3CR1$^+$CCR2$^{hi}$ monocytes accumulate early within intestinal sterile injury site and change their phenotype overtime.**
**a** Representative images taken from 6 to 96 h after focal intestinal injury and **b** quantification of their monocyte hues within injury in control *Cx3cr1$^{GFP/+}$Ccr2$^{RFP/+}$* mice. White dotted line indicates injury area. Scale bars, 100 μm. $n = 5$ per group. **c** Flow cytometry analysis of CX3CR1$^+$ and CCR2$^+$ cells in injured colon isolated from *Cx3cr1$^{GFP/+}$Ccr2$^{RFP/+}$* mice with or without burn injury (6 h). Cells were pregated on size, viability, and CD45$^+$. **d** Quantification of the proportion of CX3CR1$^-$CCR2$^+$, CX3CR1$^+$CCR2$^{hi}$, and CX3CR1$^+$CCR2$^{lo}$ cells per CD45$^+$ cells. $n = 3$ per group. **e** Quantification of the proportion of CD11b$^+$Ly6C$^+$ cells per CX3CR1$^+$CCR2$^{hi}$ cells. $n = 3$ per group. **f** Representative image of the colon 7 days after burn injury (left). High-magnification 3D image was shown in right. Scale bar, 100 μm. **g** Localization (adjacent or non- adjacent to reconstructed blood vessels) of CX3CR1$^+$ cells 7 days after burn injury. $n = 3$ per group. Data represent mean ± SEM. **$p < 0.01$, ***$p < 0.001$, NS, not significant. Source data are provided as a Source Data file.

development towards mature CX3CR1$^+$ macrophages. This contributes to injury and these monocytes are dubbed pathogenic. However, some studies suggest that classical monocytes can help repair tissue. For example, in a model of ileus, *Ccr2$^{-/-}$* mice had a worse phenotype[12] and in a model of toxoplasmosis classical monocytes had regulatory functions[37]. Therefore, we developed a simple acute thermal injury avoiding the use of chemicals that could have lingering direct effects on the immune cells and found that classical monocytes converting to macrophages were absolutely critical for timely restitution. Moreover, the environment in this type of injury was conducive to rapid conversion of CCR2$^+$ monocytes to CX3CR1$^+$ macrophages (within 3–4 days) and Nr4a1 was necessary for appropriate healing of tissue to occur, including clearance of debris and construction of mature (low permeability) vasculature. By contrast, despite a similar recruitment of monocytes in DSS colitis we found that Nr4a1 had a more minor impact on healing, consistent with the work of Bain et al., who proposed an impairment in monocyte conversion in this chemical model of intestinal inflammation[8]. Further blocking Nr4a1 did not reveal a very significant phenotype. Whether it is due to absence of a stimulatory signal or the presence of an inhibitory molecule, the monocytes failed to convert appropriately in DSS colitis to mature resident mucosal lamina propria macrophages.

The statistics for antibiotic use in pediatric medicine are staggering; 50% of human infants are exposed to antibiotics within the first year of life[38] and even a single dose of intrapartum antibiotics causes dysbiosis that persists to at least 3 months of age[39]. Our model predicts that newborns receiving antibiotics could potentially have an impaired macrophage barrier function and reduced macrophage repair capacity. Indeed, premature babies receiving antibiotics to prevent sepsis are subsequently much more susceptible to infections and necrotizing enterocolitis[40–43], a condition potentially induced by bacterial translocation and hallmarked by bleeding into the intestinal lumen.

## Methods

**Mice.** C57BL/6 mice, *Cx3cr1$^{GFP/+}$* (knock-in), *Cx3cr1$^{GFP/GFP}$* (CX3CR1-deficient), *Nr4a1$^{-/-}$*, and *LysM$^{cre}$* mice were obtained from The Jackson Laboratory. Generation of *Ccr2$^{RFP/RFP}$* (CCR2-deficient) and *Ccr2$^{RFP/+}$* (knock-in) mice have been previously described[44]. *Cx3cr1$^{GFP/+}$Ccr2$^{RFP/+}$* mice were generated by crossing *Cx3cr1$^{GFP/GFP}$Ccr2$^{RFP/RFP}$* mice with C57BL/6 mice. *Nr4a1$^{-/-}$Cx3cr1$^{GFP/+}$Ccr2$^{RFP/+}$* mice were generated by crossing *Nr4a1$^{-/-}$Cx3cr1$^{GFP/GFP}$Ccr2$^{RFP/RFP}$* mice with *Nr4a1$^{-/-}$* mice. *Myd88$^{-/-}$* and *Myd88$^{-/-}$Trif$^{-/-}$* double knockout mice were a gift from Dr. S. Akira (Osaka University, Japan). *Csf1r$^{LsL-DTR}$* mice were a gift from Dr. M. Nussenzweig. To deplete intestinal macrophages, *Csf1r$^{LsL-DTR}$* mice were crossed *LysM$^{cre}$* mice and DT (4 ng/g mouse body weight) was administered intraperitoneally 24 before the experiments. Germ-free *Cx3cr1$^{GFP/GFP}$* mice were bred and maintained in flexible film isolators at the IMC, University of Calgary, Canada. Germ-free status was confirmed by culture-dependent and -independent methods and were independently confirmed to be pathogen-free. All mice were on a C57BL/6 background. Animals were maintained in a specific pathogen-free environment with ad libitum access to food and water at the University of Calgary Animal Resource Centre. Mice were housed under standardized conditions of temperature (21–22 °C) and illumination (12/12 h light/dark cycle). Mice of 8–12 weeks of age were used for experiments. All experiments were approved by the University of Calgary Animal Care Committee and were in compliance with guidelines established by the Canadian Council for Animal Care.

**Antibodies and reagents.** Antibodies against CD3 (17A2), CD11b (M1/70), CD11c (N418), CD31 (PECAM-1, 390), CD45 (30-F11), FceR1 (MAR-1), and F4/80 (BM8) were obtained from eBioscience. Antibodies against CD80 (16-10A1), CD117 (2B8), CD206 (C068C2), CX3CR1 (SA011F11), Ly6C (HK1.4) and Ly6G (1A8) were obtained from Biolegend. Antibody against NK1.1 (PK136), CD4 (RM4-5), and CD103 (M290) was obtained from BD Pharmingen. SYTOX Orange and SYTOX Green were obtained from Thermo Fisher Scientific. TRITC-albumin was obtained from Sigma-Aldrich. For intravital imaging, we used a concentration of 2 μg per mouse of each antibody.

**In vivo treatment.** For TLR agonist treatment, mice were administered orally with 50 μg of lipoteichoic acid (LTA; Sigma-Aldrich), 50 μg of lipopolysaccharide (LPS; Sigma-Aldrich), or 50 μg of CpG (Sigma-Aldrich) once a week for 6 weeks. The dose of agonists was decided as previously described[30,31] or according to the manufacturer's instructions.

**Bacteria.** *Staphylococcus aureus* strain MW2 was kind gift from Dr. Micheal Otto (NIH/NIAID). mCherry MW2 was kind gift from Nijland R (University Medical Center Utrecht, The Netherlands). *Escherichia coli* strain XEN 14 was purchased from PerkinElmer (Norwalk, CT). *Salmonella typhimurium* strain SL1344 was kind gift from Dr. Rebekah DeVinney (University of Calgary, Canada). mCherry-*S. typhimurium* SL 1344 was generated as previously described[45]. All Gram-positive bacteria were grown BHI media, and Gram-negative bacteria were grown in LB media. The cultures were grown overnight, followed by subculture until logarithmic phase (OD 660 nm = 1). Chloramphenicol (10 μg/mL) for the pCM29 plasmid were used appropriately. To induce intestinal *S. aureus* infection, mCherry-MW2 ($1 \times 10^9$ CFU) was administered by gavage. In Abx-treated mice, Abx water was changed to normal water 2 days before the administration. To induce bacterial translocation in intestinal wall, mCherry-SL1344 ($1 \times 10^9$ CFU) was injected into 20 mm length of isolated colon. Oral and anal side of the isolated colon was ligated with 5-0 silk string and the wall was cut and imaged 6 h after the injection. For experiments with heat-killed bacteria, *S. aureus* (MW2), *E. coli* (XEN 14), and *S. typhimurium* (SL1344) were killed by incubation for 30 min at 65 °C, and heat-killed bacteria ($2 \times 10^8$ CFU) were administered orally once a week for 6 weeks.

**Bacteriological analysis.** Anesthetized mice were washed with 70% ethanol under sterile conditions. Blood was collected in a heparinized syringe by cardiac puncture. The, liver, spleen, kidney, mesenteric lymph node, and colon were removed after laparotomy, weighed, and homogenized. For determination of colony forming units (CFU), 10 μL of tissue homogenate or blood was serially diluted, plated onto BHI or LB agar plates, and incubated at 37 °C for 24 h, and then bacterial colonies were counted.

**Preparation of the mouse intestine and liver for intravital imaging.** Mice were anesthetized by s.c. injection of 200 mg/kg ketamine (Bayer Animal Health) and 10 mg/kg xylazine (Bimeda-MTC). After anesthesia, the right jugular vein was cannulated to administer fluorescent dyes and additional anesthetic. For intestinal imaging, a midline incision followed by a left lateral incision along the costal margin to the midaxillary line was performed to expose the intestine. To image the small intestinal villi or colonic lamina propria (infection or DSS colitis model), two holes were made using a cautery pen at two opposite ends of the intestine and the midline of the intestine between the two holes was cauterized down. After rinsing the mucosal side with warm saline, the cut edges of the intestine were pinched using fine-tip retractable forceps then tied with 5-0 silk strings. Mice was placed in a left lateral position and the sutures were pulled in opposite directions while normally perfused to unfold and flatten the intestine against the coverslip on the inverted microscope stage. To image the LP or the SM layer, oral side of the intestine was ligated with 5-0 silk string. PBS (100 μL/20 mm of SI or 200 μL/20 mm of colon) was introduced into the intestinal lumen using a syringe with a 30-gauge needle, and the anal side of the intestine was ligated with 5-0 silk string. Mice was placed in a left lateral position and intestine was placed on a glass coverslip and imaged if blood flow was normal. We defined "intestinal villi" as 10–30 μm of depth by imaging from mucosa. We also defined "muscularis" as 10–30 μm of depth, 'submucosa' as 30–50 μm of depth, and "lamina propria" as 50–80 μm of depth by imaging from serosa. Mice were placed on a heating pad

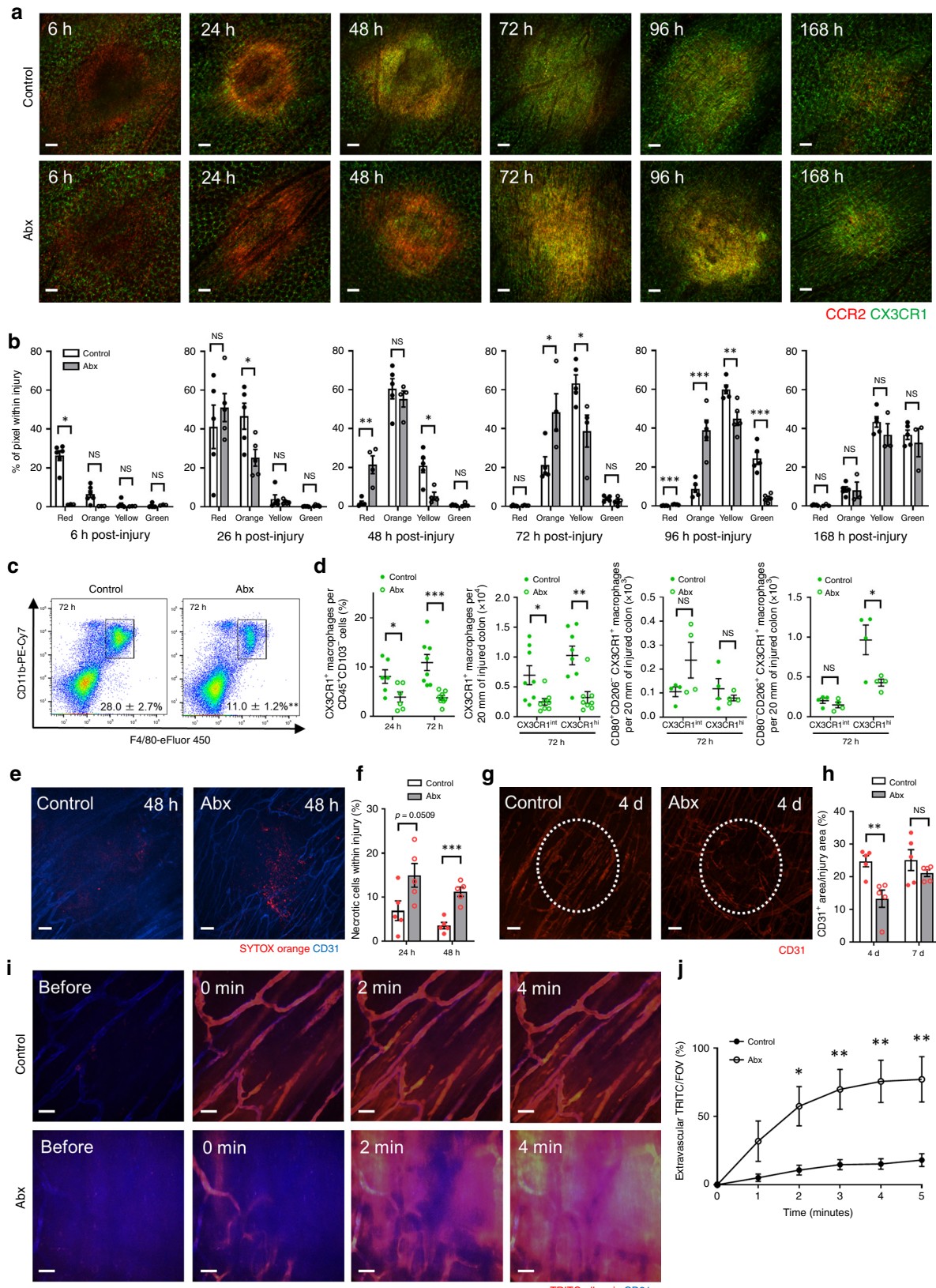

to maintain a body temperature of 37 °C throughout imaging. Exposed abdominal tissues were covered with saline-soaked gauze to prevent dehydration.

For liver imaging, mice were anesthetized and cannulated as described above, and intravital imaging was performed as previously described in ref. [46].

**Spinning disc confocal intravital microscopy (SD-IVM) and image analysis.** Multichannel spinning-disk confocal microscope was used to image mouse

intestine and liver. Image acquisition of the intestine and liver was performed using an Olympus IX81 inverted microscope (Olympus, Center Valley, PA), equipped with an Olympus focus drive and a motorized stage (Applied Scientific Instrumentation, Eugene, OR) and fitted with a motorized objective turret equipped with ×4/0.16 UPLANSAPO, ×10/0.40 UPLANSAPO, and ×20/0.70 UPLANSAPO objective lenses and coupled to a confocal light path (WaveFx; Quorum Technologies) based on a modified CSU-10 head (Yokogawa Electric Corporation,

**Fig. 7 Gut microbiota is involved in the intestinal repair process by regulating the conversion of monocytes. a** Representative images taken from 6 to 168 h after focal intestinal injury and **b** quantification of their monocyte hues within injury in control or Abx-treated $Cx3cr1^{GFP/+}Ccr2^{RFP/+}$ mice. Scale bars, 100 μm. $n = 4$–6 per group. **c** Flow cytometry analysis and **d** quantification of the proportion and number of CX3CR1$^+$ macrophages (total, CD80$^+$CD206$^-$, or CD80$^-$CD206$^+$) in injured colon isolated from control and Abx-treated mice at indicated time points after burn injury. Cells were pregated on size, viability, CD45$^+$, CD103$^-$, CD11b$^+$, and F4/80$^+$. Data are representative of three independent experiments. $n = 4$–8 per group. **e** Representative images of necrotic cells (SYTOX orange, red) within injury at 48 h post injury. Scale bars, 100 μm. **f** Quantification of SYTOX orange$^+$ area within injury at 24 and 48 h post injury in control and Abx-treated mice. $n = 5$ per group. **g** Representative images of injury site 4 days post injury in control and Abx-treated mice. Mice were administered anti-CD31 (red) antibody intravenously to visualize vasculature. White dashed line highlights original injury border. Scale bars, 100 μm. **h** Quantification of revascularization (CD31$^+$ area within injury) at indicated time points in control and Abx-treated mice. $n = 5$ per group. **i** Representative images of intestinal injury site 4 days post injury. Time-lapse images were taken before and after intravenous TRITC-albumin (red) administration in control and Abx-treated mice. Scale bars, 50 μm. **j** Quantification of extravascular (outside of CD31$^+$ vasculature) TRITC$^+$ area/FOV at indicated time points in control and Abx-treated mice. $n = 5$ per group. Data represent mean ± SEM. $*p < 0.05$, $**p < 0.01$, $***p < 0.001$, NS not significant. Source data are provided as a Source Data file.

Tokyo, Japan). Cells of interest were visualized using fluorescently labeled antibodies, and fluorescent reporter mice. In some experiments, necrotic cells were labeled by superfusion of the intestinal surface with 1 μM SYTOX Orange or SYTOX Green. Laser excitation wavelengths of 491, 561, 642, and 730 nm (Cobolt, AB, Solna, Sweeden) were used in a rapid succession together with the appropriate band-pass filters (Semrock Inc., Rochester, NY). A back-thinned electron-multiplying charge-coupled device 512 × 512-pixel camera (Hamamatsu Photonics) was used for fluorescence detection. Volocity software 6.1 (PerkinElmer) was used to drive the confocal microscope and for 3D rendering, acquisition, and analysis of images.

Acquired images were analyzed or exported as TIF images using Volocity software. The minimum threshold values were adjusted for each of the fluorescence channels to reduce background. Exported images were imported to Image J software package (NIH) for analysis[47]. For quantification of the area of GFP$^+$ cells and RFP$^+$ cells in the intestine, a total of 8–10 images of the indicated layer were acquired for each mouse using a ×10 objective and was evaluated using the 'analyze and measure' command in Image J software. The data obtained from each image were averaged and counted $n = 1$. The area where the intestinal tissue does not exit was excluded from the calculation. Area of the vasculature covered by CX3CR1$^+$ cells was quantified by comparing TIFF images holding the blue channel (CD31-APC) and green channel (CX3CR1-GFP) and the same TIFF images holding only the blue channel. The GFP-unmixed CD31$^+$ area was measured by image J and covered area was calculated. A total of 3–5 images were acquired for each mouse using a ×20 objective and the data obtained from each image were averaged and counted $n = 1$. Monocyte hues were measured as previously described in ref. [48]. Briefly, RGB TIF images (where the blue channel was blank) were imported to Image J. Images were segmented using the "Color Threshold Tool" in the brightness channel in the hue, saturation and brightness (HSB) color space. Threshold amounts for each dataset were identical in the same imaging session. Pixels in the segmented color region were categorized according to the hue (red, orange, yellow or green) before the relative percentage of pixels in each color category was calculated. Quantification of SYTOX Orange/Green positive dead cells in the intestinal injury area was performed using Image J as previously described in refs. [47,49]. Autofluorescence induced by debris was excluded from the analysis.

**Depletion of gut commensal bacteria.** Gut commensal bacteria were depleted using a method modified a protocol as previously described[27]. Mice were provided with ampicillin (1 g/L), vancomycin (0.5 g/L), neomycin (1 g/L), metronidazole (1 g/L), and ciprofloxacin (0.2 g/L) in drinking water. All antibiotics were obtained from Sigma-Aldrich. Antibiotics-mixed water was started from E14.5 and continued until the experiments. In bone marrow transplant experiments, antibiotics-mixed water was started just after the transplantation.

**SYTOX green staining of intestinal content.** Bacterial load of intestinal contents was measured as previously described in ref. [50]. Briefly, a fecal pellet was collected from a control or an Abx-treated mouse and each sample was added 500 μL 4% paraformaldehyde and mixed. After the incubation for 30 minutes at room temperature, 100 μL of fixed content was resuspended in 800 μL sterile PBS and 1 μL of SYTOX green (100 μg/mL) was added. After the incubation for 30 min in the dark, samples were centrifuged and 50 μL of supernatant was spread on slide. The image was recorded by microscopy in green fluorescence channel using a ×10 objective. A total of 4 images were acquired for each mouse and the number of SYTOX green$^+$ particles was counted using Image J software. Data obtained from each image were averaged and counted $n = 1$.

**Generation of bone marrow chimeric mice.** Bone marrow chimeric mice were generated by bone marrow transplantation using a standard protocol as previously described[51]. Wild-type C57BL/6, $MyD88^{-/-}$, or $Myd88^{-/-}Trif^{-/-}$ mice were used

as recipients, and C57BL/6, $Cx3cr1^{GFP/+}Ccr2^{RFP/+}$, $Cx3cr1^{GFP/+}Ccr2^{RFP/RFP}$, $Nr4a1^{-/-}Cx3cr1^{GFP/+}Ccr2^{RFP/+}$, $MyD88^{-/-}$, or $Myd88^{-/-}Trif^{-/-}$ mice were used as donors. Bone marrow was isolated from donor mice euthanized by cervical spine dislocation. Recipients were irradiated with two doses of 5 Gy (Gammacell 40 $^{137}$Cs γ-radiation source), with a 3-h interval between doses. Thereafter, $6 × 10^6$ isolated donor bone marrow cells were injected via the tail vein into irradiated recipient mice. Transplanted mice received 0.2% neomycin water for 2 weeks or antibiotics-mixed water until experiments, and remained in germ-free microisolator cages.

**Sterile inflammation induced by focal necrotic injury.** Mice were anesthetized with isoflurane and a small midline laparotomy was made to exteriorize the colon. For imaging, a single focal injury was induced on serosal surface of the colon to a depth of 80 μm using the tip of a heated 30-gauge needle mounted on an electrocautery device. Meanwhile, for analysis of flow cytometry, 10 points of injury per 20 mm of colon were induced. The incision was sutured closed and mice were allowed to recover for imaging or flow cytometry analysis of later time points (6, 24, 48, 72, 96, 168 h) after injury. For sham experiments, mice underwent the same surgical procedure, but no injury was induced.

**Isolation of colonic lamina propria cells.** LP cells were extracted from the total colon for analysis under basal conditions. Meanwhile, when analyzing the injured colon, LP cells were extracted from 20 mm of colon with 10 points of injury. Colon was washed with PBS after opened longitudinally and cut into small pieces. They were incubated with PBS containing 3 mM EDTA for 15 min and RPMI-1640 medium containing 1% fetal calf serum, 1% penicillin-streptomycin, 1 mM EGTA, and 2 mM MgCl$_2$ for 20 min at 37 °C each repeatedly twice. Intestinal lamina propria cells were isolated by digestion with 0.5 mg/mL Collagenase Type XI (Sigma), 0.5 mg/mL Dispase II (Roche) and 50 μg/mL DNase I (Sigma) for 60 min at 37 °C (RPMI-1640 medium containing 10% fetal calf serum and 1% penicillin-streptomycin). They were additionally incubated for 30 min after homogenized (18G needle and 10 mL syringe). Thereafter they were homogenized again and pushed through a 40 μm filter. For intracellular cytokine staining, samples were placed into PBS containing 3.0 μg/mL brefeldin A solution for 1 h at 37 °C under 5% CO2 condition immediately after the cutting of samples to inhibit intracellular protein transport.

**Flow cytometry.** After mice were anesthetized, blood and femur were collected and placed in PBS on ice. Single-cell suspensions of bone marrow were generated by mechanical disruption through a 40-μm nylon mesh (BD Bioscience). Colonic LP cells were isolated as described. Residual red blood cells were lysed using ACK lysing buffer (Invitrogen). The cells were blocked using anti-CD16/32 antibody (2.4G2 clone; Bio X Cell) for 30 min. Then, cells were stained for 30 min with antibodies for specified markers. Nonviable cells were identified using viability dye efluor 780 (eBioscience) or Ghost Dye$^{TM}$ Red 710 (TONBO biosciences). Samples were run using a flow cytometer (FACSCanto; BD Biosciences, LSR-II; BD Biosciences) and analyzed using FlowJo software (Tree Star).

**Vascular permeability measurement.** Time-lapse imaging of the colon was recorded within injury area 4 days post injury. One minute after the start of imaging, TRITC or FITC-albumin was administered intravenously (0.5 mg/mouse) and time-lapse imaging was recorded for 5 min thereafter. Extravascular (outside of CD31$^+$ vasculature) TRITC$^+$ or FITC$^+$ area/FOV was quantified using Image J software every 1 min in WT control, $Ccr2^{RFP/RFP}$, $Nr4a1^{-/-}$, and Abx-treated mice.

**Measurement of colon tissue cytokine concentration.** Quantification of concentrations of cytokines was performed using the validated Luminex bead-based assay from R&D Systems (Minneapolis, MN) according to manufacturer's instructions. Briefly, a 20-mm-long colon tissue sample was placed into 1 mL PBS,

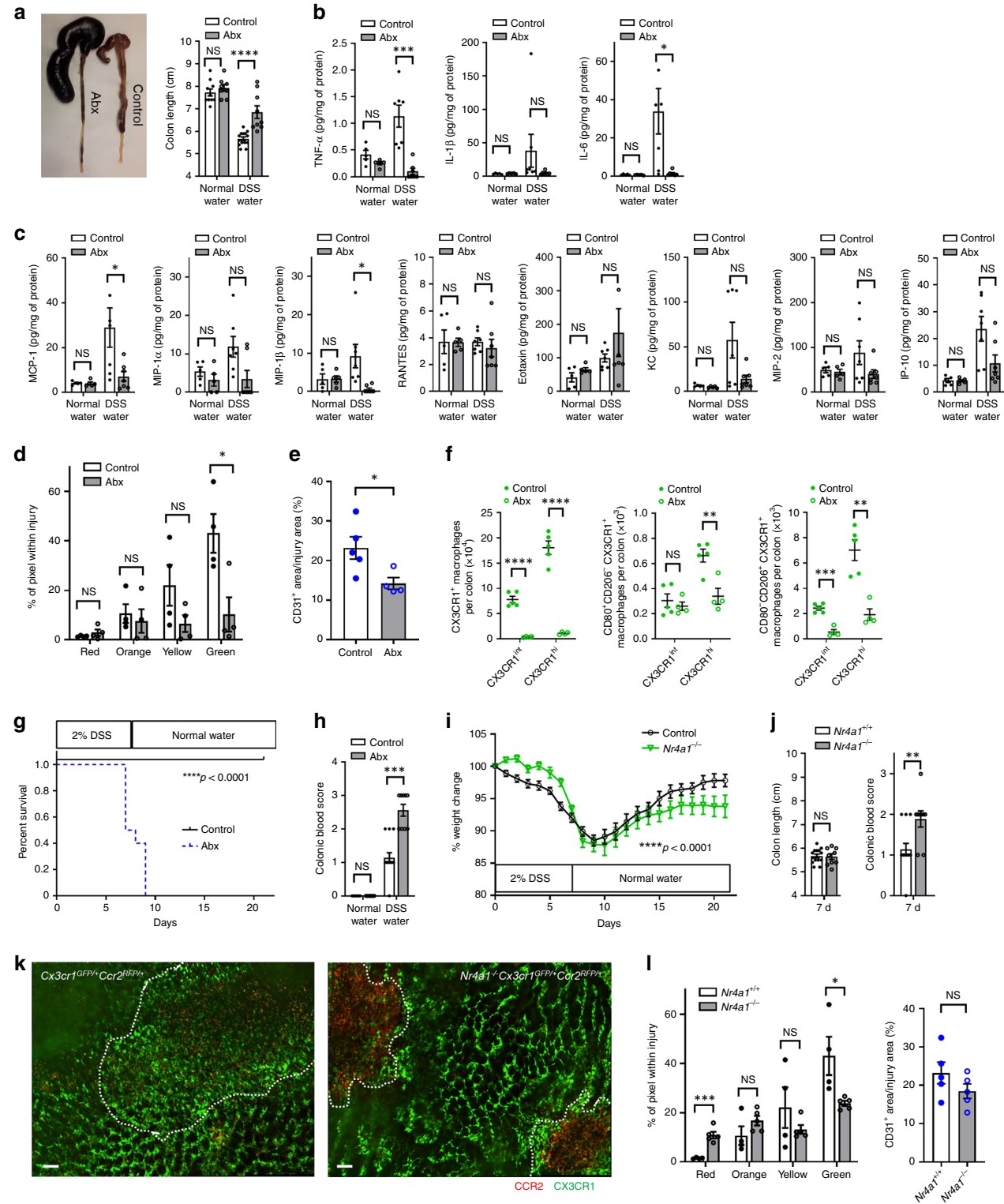

and they were mechanically disrupted. The mixture was then centrifuged at 10,000 rpm for 5 min three times and the supernatant was transferred to a 0.22 μm PVDF DuraPore centrifugal filter (EMD Millipore, Billerica, MA) to remove any particles from the solution. Filtered samples were then incubated with capture bead cocktail on a 96-well plate in the dark for 2 h at room temperature. After incubation, wells were washed with wash buffer, incubated with a biotin antibody cocktail for 1 h next before another round washing and incubation for another 30 min with Streptavidin-PE. Following washes, the plate was read using a Luminex 200 apparatus (Applied Cytometry Systems, UK) and analyzed with StarStation V.2.3 (Applied Cytometry Systems, UK). Total protein concentration of filtered samples

was measured using Bio-Rad Protein Assay (Bio-Rad Laboratories). Results were normalized against the amount of total protein extracted from the colon tissues.

**DSS colitis model.** Mice were given 2.0% DSS (molecular mass, 36,000–50,000, MP Biomedicals, LLC) in the drinking water for continuous 7 days, then followed with normal water for another 14 days. Survival was observed daily in wild-type control and Abx-treated mice. Body weight was observed daily in wild-type control and $Nr4a1^{-/-}$ mice. At 7 days after the start of DSS water, colon length in control, $Nr4a1^{-/-}$, and Abx-treated mice was measured, and the colonic blood score[30] was

**Fig. 8 Gut microbiota affects the tissue repair process in DSS colitis. a** Macroscopic findings of the colon 7 days after the start of DSS water in control and Abx-treated mice (left) and quantification of colon length (right; $n = 8$–14 per group). Luminex assays of **b** pro-inflammatory cytokines, **c** chemokines in colon tissue samples at steady state and at 7 days of DSS-treatment ($n = 5$–7 per group). Quantification of **d** monocyte hues and **e** CD31$^+$ area within injury site of lamina propria in control and Abx-treated $Cx3cr1^{GFP/+}Ccr2^{RFP/+}$ mice at 7 days of DSS-treatment. $n = 3$–5 per group. **f** Quantification of the number of CX3CR1$^+$ macrophages (total, CD80$^+$CD206$^-$, or CD80$^-$CD206$^+$) in colon isolated from control and Abx-treated mice after 7 days of DSS treatment. Cells were pregated on size, viability, CD45$^+$, CD103$^-$, CD11b$^+$, and F4/80$^+$. $n = 4$–5 per group. **g** Survival of control and Abx-treated mice in 2% DSS-treatment. $n = 10$ per group. Data were pooled from two independent experiments. **h** Colonic blood score 7 days after the start of DSS water ($n = 8$–14 per group). **i** Body weight change of control and $Nr4a1^{-/-}$ mice in 2% DSS colitis. $n = 10$ per group. **j** Quantification of colon length and colonic blood score ($n = 9$–14 per group). **k** Representative stitch images of lamina propria and **l** quantification of monocyte hues and CD31$^+$ area/injury area in $Cx3cr1^{GFP/+}Ccr2^{RFP/+}$ and $Nr4a1^{-/-}Cx3cr1^{GFP/+}Ccr2^{RFP/+}$ mice at 7 days of DSS-treatment. White dotted line indicates injury area. Scale bars, 100 μm. $n = 3$–5 per group. Data represent mean ± SEM. *$p < 0.05$, **$p < 0.01$, ***$p < 0.001$, ****$p < 0.0001$, NS not significant. Source data are provided as a Source Data file.

evaluated as follows; 0 = lack of any gross blood visible in the colon, 1 = gross blood present in <1/3 of the colon, 2 = gross blood present in <2/3 of the colon, 3 = gross blood present in >2/3 of the colon. Mucosal surface of the colon in control and Abx-treated $Cx3cr1^{GFP/+}Ccr2^{RFP/+}$ mice or $Nr4a1^{-/-}Cx3cr1^{GFP/+}Ccr2^{RFP/+}$ mice was imaged at 7 days after the start of DSS water using SD-IVM. In some experiments for *S. aureus* infection, mice were given 4.0% DSS water for 5 days to induce acute DSS colitis.

**Statistical analysis**. Data were expressed as mean ± SEM. Unpaired Student *t* test was used to compare between two groups. Bacterial CFU data and colonic blood score were compared by Mann–Whitney *U* test. One-way ANOVA was used to compare more than two groups, followed by Tukey's post hoc test. Survival rate was evaluated by the Kaplan–Meier method, and the log-rank test was used to compare survival curves. Percent body weight change in WT and $Nr4a1^{-/-}$ mice in 2% DSS-treatment were compared by two-way repeated-measures ANOVA. A *p* value < 0.05 was considered statistically significant. All tests were two-tailed. All statistical analyses were performed using GraphPad Prism v7.0 software (GraphPad Software Inc., La Jolla, CA).

**Reporting summary**. Further information on research design is available in the Nature Research Reporting Summary linked to this article.

## Data availability

We declare that the data supporting the findings of this study are available within the paper and its supplementary information file. The source data underlying Figs. 1c, e, g, 2b, e, g, h, 3c, d, g, I, 4b–f, 5b, c, g, k, 6b, d, e, g, 7b–d, f, h, j, 8a–j, l, and Supplementary Figs. 1b, 2b, c, 3b, d, 4b, d, 5a–d, 6b, d, f, g, 7b, e, 8b, c, e–h are provided as Source Data file.

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

## Acknowledgements

We thank Trecia Nussbaumer for mice husbandry. We thank Karen Poon from the Snyder Institute Molecular Core for assistance with flow cytometry. This work is supported in part by the International Microbiome Centre (IMC) funded through the Cumming School of Medicine, University of Calgary, Western Economic Diversification (WED) and Alberta Economic Development and Trade (AEDT), Canada. M.H. is supported by the Research Fellowship of the Uehara Memorial Foundation and grants from the Ministry of Education, Culture, Sports, Sciences and Technology of Japan (KAKENHI 19H03716). M.W. is supported by the Postdoctoral Fellowship of the Uehara Memorial Foundation. B.G.S. is supported by the Canadian Institutes of Health Research (CIHR). P.K.'s laboratory is supported by a foundation grant from the CIHR, Alberta Innovates Health Solutions, the Heart and Stroke Foundation of Canada and the Canada Research Chairs program.

## Author contributions

M.H. designed and performed experiments, analyzed data and wrote the paper.t. B.S. generated mCherry-*S. typhimurium* and performed experiments, analyzed data and wrote the manuscript. M.W. helped with flow cytometry experiments. H.C. contributed to the critical review of the paper. W.L. performed bone marrow transplantation experiments. K.B. and K.M prepared and performed germ-free mice experiments. C.C.H. provided the NR4A1$^{-/-}$ mice and advice. P.K. provided overall supervision and wrote the paper.

## Competing interests

The authors declare no competing interests.
