## [Peer Review File · Nature Communications]

Editorial Note: This manuscript has been previously reviewed at another journal that is not operating a transparent peer review scheme. This document only contains reviewer comments and rebuttal letters for versions considered at Nature Communications. Mentions of prior referee reports have been redacted.

Reviewer #3:

Remarks to the Author:

In this manuscript, Honda et al. describe how macrophages in the lamina propria of the intestine associate intimately with blood vessels to act as a perivascular barrier. They use microscopy to visualise monocytes and macrophages during different models of intestinal injury. Many of their findings confirm previous data, although the spatial positioning of mucosal macrophages in relation to blood vessels is novel and will be of value to the field. However, there are many times where the authors overinterpret their data and there are points that need to be addressed.

Major points:

1. One major issue with the system employed by the authors is that their microscopy often isn't and cannot be validated using flow cytometry. The Ccr2RFP x Cx3cr1GFP mouse should allow the tracking of monocyte to macrophage differentiation in the gut (from RFP+GFP- monocytes to RFP-GFP+ macrophages through a series of intermediaries). However, while this is reported by the authors to occur in their imaging studies, flow cytometry of this double reporter mouse shows that all CCR2+ cells (RFP+) myeloid cells are CX3CR1hi (i.e. GFP^{hi}). The RFP^{lo} cells, that is the cells that have presumably lost CCR2, also express lower levels of GFP (i.e. are CX3CR1^{intermediate}) – this is illustrated in Supplementary Figure 1e and Figure 6c. Nowhere in the manuscript do the authors address this inconsistency. How can the authors be confident that their microscopy studies are faithfully capturing in situ monocyte differentiation when they cannot validate this using a complementary technique. Indeed, one criticism of this paper is that the authors are very selective when they show validation by flow cytometry, often choosing a select time point to show flow data. The manuscript would benefit from a more transparent description of the system used and the potential limitations. Plus complementary flow data should be shown for each system as a confirmation of their findings. This is vital for proper interpretation of the data and to prevent misinterpretation by other authors using these tools. The fact the flow and microscopy does not match must be addressed in the Discussion.
2. How can the authors rule out that the red cells seen in Figures 1d and f are not CCR2+ T cells referred to in Lines 148-152? They show in Supplementary Figure 1e that there is a significant fraction of T cells amongst the RFP+ cells. What criteria has been used to distinguish between monocytes and T cells? This should be stated.
3. The use of CD80 and CD206 is rather meaningless and it is not justified to refer to the 20% of macrophages expressing CD206 as “anti-inflammatory” (lines 126-128). The authors have provided no data to substantiate this claim. If anything, the authors have identified heterogeneity on the basis of CD206.
4. On lines 142-145, the authors state that “imaging revealed an orange/yellow appearance” of monocytes. I could not see data to support this statement.
5. The authors use sublethal irradiation and BMT to assess the repopulation kinetics of gut macrophages. The reduced ability of CCR2 KO-derived cells to repopulate the gut macrophage compartment is completely expected given the work of the Malissen, Jung and Mowat groups. The authors then overinterpret their data by stating that reduced repopulation led to a “significant reduction in elongated green macrophages suggesting reduced barrier function”. There are no data at this point to support this claim. A similar criticism could be made of the Nr4a1 KO data. The authors show that macrophages are unaffected in terms of abundance in intact Nr4a1 KO mice (S. Figure 8c) but that BMT of Nr4a1 BM leads to reduced macrophage numbers. The authors should acknowledge this difference and discuss why this might be.

6. On line 203, the authors should use the word 'influence' rather than 'help' when referring to the microbiota. Help infers a positive effect and the authors are simply testing the hypothesis that the microbiota alters recruitment/differentiation in some way.

7. The authors state that the vasculature is unaffected by GF conditions. The image shown does not support this statement. If gnotobiotic conditions has no effect on the vasculature, then the authors should show a more representative image.

8. Inclusion of CD80/CD206 in Figure 3i is uninformative.

9. The authors use MyD88 and MyD88/TRIF KO BM to assess the role of signalling through the TLR pathways in monocyte maturation. They show that both KO strains are able to replenish the irradiated niche and then conclude that "These findings raise the possibility that TLRs and MyD88 or MyD88/Trif signaling are not essential in the recruitment, localization and differentiation of intestinal monocytes/macrophages in response to bacterial products under homeostatic conditions." This is somewhat of an overinterpretation. These are by no means homeostatic conditions. This is a massive repopulation event. This should be re-written to reflect this. Also, the authors should substitute the word differentiation for survival. They have not performed any experiments to assess the differentiation status of these cells. While this may seem like semantics, recent work has gone to great lengths to look into tissue/niche-specific imprinting of macrophage differentiation. This study has not considered the transcriptome/proteome of macrophages in different contexts, so they cannot make this conclusion.

10. In Figure 4e, the effects of ABX cannot be determined because the authors do not include any controls (without ABX). Therefore, these data do not support the conclusions of the authors. Similarly, Figure 5d has no control image making interpretation difficult.

11. The LysM-CrexCsf1r-DTR system is a nice way to have increased specificity in terms of depletion compared with e.g. CD11c-DTR mice. However, despite causing >75% reduction in macrophages in this system, this appears to have no effect on the bacteria in the colon and a relatively small, albeit significant, effect on the dissemination to liver/spleen. The figure legend does not state how many times this experiment was performed. Given importance of this finding, it is important to state the reproducibility of this finding. Please add this.

12. Flow cytometry data should be shown for the entire time course in Figure 6. Again, as described above, I cannot understand why the flow cytometry data do not match the imaging data here.

Response to the new Reviewer #3

In this manuscript, Honda et al. describe how macrophages in the lamina propria of the intestine associate intimately with blood vessels to act as a perivascular barrier. They use microscopy to visualise monocytes and macrophages during different models of intestinal injury. Many of their findings confirm previous data, although the spatial positioning of mucosal macrophages in relation to blood vessels is novel and will be of value to the field. However, there are many times where the authors overinterpret their data and there are points that need to be addressed.

Thank you very much for your review of our manuscript. We have taken what you have pointed out and we have completely re-written and revised the manuscript and removed many many of the interpretations.

Major points:

1) One major issue with the system employed by the authors is that their microscopy often isn't and cannot be validated using flow cytometry. The Ccr2RFP x Cx3cr1GFP mouse should allow the tracking of monocyte to macrophage differentiation in the gut (from RFP+GFP- monocytes to RFP-GFP+ macrophages through a series of intermediaries). However, while this is reported by the authors to occur in their imaging studies, flow cytometry of this double reporter mouse shows that all CCR2+ cells (RFP+) myeloid cells are CX3CR1hi (i.e. GFP+). The RFPlo cells, that is the cells that have presumably lost CCR2, also express lower levels of GFP (i.e. are CX3CR1intermediate) – this is illustrated in Supplementary Figure 1e and Figure 6c. Nowhere in the manuscript do the authors address this inconsistency. How can the authors be confident that their microscopy studies are faithfully capturing in situ monocyte differentiation when they cannot validate this using a complementary technique. Indeed, one criticism of this paper is that the authors are very selective when they show validation by flow cytometry, often choosing a select time point to show flow data. The manuscript would benefit from a more transparent description of the system used and the potential limitations. Plus complementary flow data should be shown for each system as a confirmation of their findings. This is vital for proper interpretation of the data and to prevent misinterpretation by other authors using these tools. The fact the flow and microscopy does not match must be addressed in the Discussion.

We have now addressed some of this work in the discussion. Please note this is the 4th iteration of this manuscript and things have been put in and pulled out to satisfy reviewers. Initially, the entire study was imaging and we did some flow cytometry to validate the conclusions and they still hold. As you point out, there is a ratio in imaging of hi red cells (Ly6C hi) which appear to mask the green cells (Ly6C lo cells) so in imaging the cells are red while on flow they appear in the upper right hand corner (as has been shown by many). As the red disappears the cells become more orange yellow and green. Similar in flow they also become less red and retain their GFP. This is a log scale so changes are very subtle but in imaging they are linear. In imaging where the GFP/RFP ratio gives the impression GFP is going way up and RFP is going way down. Based on the flow cytometry the GFP looks like it does not. As such we have now addressed this by calling the classical monocytes CCR2hiCX3CR1+ and the changing to macrophage as CX3CR1+ CCR2lo to indicate that the RFP signal is changing much more than the GFP signal. Nevertheless, I think it is safe to say that there is a waterfall effect of Ly6Chi monocytes becoming Ly6Clo monocytes and ultimately macrophages using imaging that has not been performed previously and validates the work of others. Numerous groups have shown this and we were criticized in earlier versions that this was already known and that the only new data is the imaging. Our imaging indeed

recapitulates nicely the waterfall effect. The flow cytometry data was requested at certain time point and we provided it. I have added a brief statement about this flow cytometry versus imaging results.

Regardless, as you point out this is not the key novel finding. The key novel finding which has nothing to do with these transitions is the localization of the macrophage perivascularly and we have now provided this as the main new idea in this paper.

2) How can the authors rule out that the red cells seen in Figures 1d and f are not CCR2+ T cells referred to in Lines 148-152? They show in Supplementary Figure 1e that there is a significant fraction of T cells amongst the RFP+ cells. What criteria has been used to distinguish between monocytes and T cells? This should be stated.

We now state that indeed some of the red cells are T cells but they are negative in the GFP channel and never convert over to orange, yellow and green phenotype, nor do they upregulate F4/80. Again this has been stated.

3) The use of CD80 and CD206 is rather meaningless and it is not justified to refer to the 20% of macrophages expressing CD206 as “anti-inflammatory” (lines 126-128). The authors have provided no data to substantiate this claim. If anything, the authors have identified heterogeneity on the basis of CD206.

We analyzed the macrophages and added exactly this data in response to a specific and explicit comment from Reviewer #2 in a previous revision. Several studies have reported CD80 as M1 marker and CD206 as M2 marker including intestinal macrophages (Chen et al. Cancer Immunol Immunother 2018, Nawaz et al. Nat Commun 2017, Cosin-Roger et al. PLoS One 2013), as such we measured these markers. We agree with reviewer #3's suggestion and removed the word 'anti-inflammatory'. While this reviewer feels CD80 and CD206 are meaningless, reviewer #2 felt it was essential. I would recommend that we let readers decide whether it is indeed meaningless or crucial to the story. We will remove antiinflammatory and not try to implicate a phenotype based on these markers. Thank you.

4) On lines 142-145, the authors state that “imaging revealed an orange/yellow appearance” of monocytes. I could not see data to support this statement.

Thank you for this. We added the arrows in Fig. 1d to show the orange/yellow cells.

5) The authors use sublethal irradiation and BMT to assess the repopulation kinetics of gut macrophages. The reduced ability of CCR2 KO-derived cells to repopulate the gut macrophage compartment is completely expected given the work of the Malissen, Jung and Mowat groups. The authors then overinterpret their data by stating that reduced repopulation led to a “significant reduction in elongated green macrophages suggesting reduced barrier function”. There are no data at this point to support this claim. A similar criticism could be made of the Nr4a1 KO data. The authors show that macrophages are unaffected in terms of abundance in intact Nr4a1 KO mice (S. Figure 8c) but that BMT of Nr4a1 BM leads to reduced macrophage numbers. The authors should acknowledge this difference and discuss why this might be.

Thank you for this comment. We removed the phrase 'suggesting reduced barrier function' in line 176. Our data suggest that Nr4a1 is important in ensuring that the macrophages are M2 or repair like and that in the absence of this molecule, they either fail to fully mature or fail to become a repair

macrophage. We have included both of these possibilities. Yes the Nr4a1^{-/-} mice have the same amount of macrophages under baseline suggesting other factors including the microbiome may help to mature them. We state this and highlight that in emergency repopulation is where we actually do see a phenotype as the Nr4a1^{-/-} mice struggle to repair the injured intestine (Supplemental Fig. 8). Our data do not agree with the cited study that DSS colitis in Nr4a1^{-/-} mice had increased inflammation via NF-κB in macrophages. We actually see more subtle differences with no evidence of increased inflammation, ie., the colon length is the same, and there is more bleeding but other than that there are no other differences. Bain et al., suggested that in DSS there is an interruption in monocyte conversion and our data would support this revealing little additional phenotype in the Nr4a1^{-/-} mice.

6) On line 203, the authors should use the word 'influence' rather than 'help' when referring to the microbiota. Help infers a positive effect and the authors are simply testing the hypothesis that the microbiota alters recruitment/differentiation in some way.

Thank you for this comment. We rewrote the word 'help' to 'influence'.

7) The authors state that the vasculature is unaffected by GF conditions. The image shown does not support this statement. If gnotobiotic conditions has no effect on the vasculature, then the authors should show a more representative image.

We apologize for confusing you. It is due to the technical aspects associated with 3D reconstruction. Germ-free mice did not have altered development of the intestinal vasculature at steady state (Fig. 3g). Having said that, to avoid confusion, we replaced the 3D image in Figure 3g with a much better one. Our apologies.

8) Inclusion of CD80/CD206 in Figure 3i is uninformative.

As described, we analyzed CD80/CD206 expression as a marker of macrophage phenotype in response to comment from Reviewer #2 in a previous revision. We identified that the number of CD80-CD206+ CX3CR1^{hi} macrophages in colonic LP at 72 hrs after burn injury is smaller in Abx-treated mice compared to SPF control (Fig. 7d). Since reviewer #2 finds it informative and critical and you do not, we have decided to include it and not speculate about what it means. Readers can make up their own minds.

9) The authors use MyD88 and MyD88/TRIF KO BM to assess the role of signalling through the TLR pathways in monocyte maturation. They show that both KO strains are able to replenish the irradiated niche and then conclude that "These findings raise the possibility that TLRs and MyD88 or MyD88/Trif signaling are not essential in the recruitment, localization and differentiation of intestinal monocytes/macrophages in response to bacterial products under homeostatic conditions." This is somewhat of an overinterpretation. These are by no means homeostatic conditions. This is a massive repopulation event. This should be re-written to reflect this. Also, the authors should substitute the word differentiation for survival. They have not performed any experiments to assess the differentiation status of these cells. While this may seem like semantics, recent work has gone to great lengths to look into tissue/niche-specific imprinting of macrophage differentiation. This study has not considered the transcriptome/proteome of macrophages in different contexts, so they cannot make this conclusion.

Thank you for this. We have toned down this to remove any overinterpretation. Although we document that the irradiation only affected the bone marrow and did not affect the gut macrophages (Supplementary Fig.2) you are correct that any irradiation could do something untoward. We therefore

simply state now that replenishment under these BMT conditions was not affected by the different TLR mutant animals.

10) In Figure 4e, the effects of ABX cannot be determined because the authors do not include any controls (without ABX). Therefore, these data do not support the conclusions of the authors. Similarly, Figure 5d has no control image making interpretation difficult.

Thank you for this comment. We provided the control data (with normal water) in Figure 4e. We also added the control image in Figure 5d.

11) The LysM-Crex^{Csf1r}-DTR system is a nice way to have increased specificity in terms of depletion compared with e.g. CD11c-DTR mice. However, despite causing >75% reduction in macrophages in this system, this appears to have no effect on the bacteria in the colon and a relatively small, albeit significant, effect on the dissemination to liver/spleen. The figure legend does not state how many times this experiment was performed. Given importance of this finding, it is important to state the reproducibility of this finding. Please add this.

Thank you for this comment. The effect on the dissemination to liver/spleen is statistically significant and the difference is a log scale. Data were pooled from two independent experiments where multiple animals were used. We added this statement in Figure legend.

12) Flow cytometry data should be shown for the entire time course in Figure 6. Again, as described above, I cannot understand why the flow cytometry data do not match the imaging data here.

As we explain above we do feel they match previous work and our work but there are nuanced differences between the imaging and the flow cytometry which we explain above and have no changed throughout the text to better reflect the flow cytometry and imaging data according to your suggestion in the first point.

Reviewers' comments:

Reviewer #3 (Remarks to the Author):

The authors have addressed all my comments.

Reviewer #4 (Remarks to the Author):

In this manuscript the authors show that intestinal macrophages in the lamina propria are in close proximity with the blood vessels where they create a sort of additional barrier that the authors claim to protect the vessels.

While the idea is intriguing, I am not sure that the authors have clearly demonstrated this. Further, the finding that macrophages embrace blood vessels is not new (Chang et al, Immunity 2013).

I have several concerns.

The interpretation of the data often is speculative. Some examples:

Line 147 : I don't understand how can the authors claim from the IF or flow cytometry that: 'From the flow cytometry data, it appears that it is more a loss of CCR2 and associated RFP signal than a change in CX3CR1 and a GFP signal that constitutes the red, orange, yellow and green colors.'

Line 166: Again this sentence is not supported by the data: These reporter monocytes in colonic mucosal lamina propria changed from red to orange to yellow to green, signifying a change of phenotype from CCR2^{hi}CX3CR1⁺ to CX3CR1⁺CCR2^{lo} monocytes (Fig. 2a, b).. It could just be proliferation of green monocytes that have never been red. Fig 2 f, g may suggest that these cells require CCR2 to seed the gut, or that there is cooperation between CCR2 and CX3CR1 cells to allow the latter to seed the perivascular areas. How can the authors make sure that it is not just inability of expressing CX3CR1? Can they follow CD11b⁺F4/80⁺ cells to make sure that these cells are missing?

Major concerns

Fig. 3f. I wonder how the authors can claim this is a barrier without any functional data. The endothelium itself in the gut is organized in a barrier and includes pericytes and enteric glial cells. Functional studies have shown this (Spadoni et al. Science 2015; Mouries et al. J. Hep 2019).

Fig. 3g. The vasculature has been shown to be underdeveloped in germ free mice (Reinhardt C, et al. Nature 2012). Also in this image the vasculature seems not properly developed.

Fig. 5. The experiments of pathogen dissemination in antibiotic treated mice may be explained by several other possibilities:

1. Mice lacking the microbiota are more affected by pathogens because the microbiota releases antimicrobial peptides
2. The microbiota induces IgA responses. IgAs quickly go down in the absence of the microbiota and hence also this protection is lost.
3. The microbiota induces mucus production and hence also this barrier is lost.
4. Antibiotic treatment impairs the formation of a vascular barrier and thus this is more permeable to bacteria (Mouries et al. J Hep 2019).

That macrophages readily internalize bacteria is a known function.

Fig 5K. The increased dissemination in mice lacking macrophages is quite expected as they play a major role anyway in microbial elimination. I am quite surprised that there was no difference in the gut even though the authors analyzed only the colon which is not the best site for Salmonella infection (they should have analyzed also the small intestine).

Fig. 6a. Probably the most convincing experiment of monocyte recruitment was done with thermal ablation. Here it is clear that CCR2 are recruited and probably transform into CX3CR1 (even though the clear demonstration of a transformation of these cells from CCR2+ to CX3CR1+ cells would come using a tracer system like ROSA26. Again the authors cannot exclude that CCR2+ cells favor the recruitment of CX3CR1+ cells. Indeed in CCR2-/- mice CX3CR1+ cells eventually seed the perivascular region, suggesting that they can arrive there even in the absence of CCR2.

Fig. 8. The experiments of DSS in abx treated mice are somehow misleading as germ free mice succumb because they have defects in the epithelial barrier and not because of higher inflammation (Hernández-Chirlique C et al J Crohns Colitis. 2016) Indeed they lack the bacteria responsible for the induction of the inflammatory response in the presence of DSS. This may have nothing to do with the macrophages lining the vessels.

In conclusion, I think the authors neglected a series of studies showing the existence of a gut vascular barrier that might explain most of the results observed as it is lost during abx treatment. Abx treatment also profoundly affects the mucus layer, IgA and antimicrobial peptide production. The results in Nr4A1-/- mice would rather go in this direction showing no clear effect of the reduction of macrophage coverage during DSS colitis. While it is interesting the coverage of the vasculature by CX3CR1+ cells their role in this manuscript has not been elucidated. They might favor the establishment of the gut vascular barrier.

Response to the Reviewer #4

In this manuscript the authors show that intestinal macrophages in the lamina propria are in close proximity with the blood vessels where they create a sort of additional barrier that the authors claim to protect the vessels.

While the idea is intriguing, I am not sure that the authors have clearly demonstrated this. Further, the finding that macrophages embrace blood vessels is not new (Chang et al, Immunity 2013).

Thank you for your review. I just want to make sure we are on the same page, we suggest that the macrophage are a barrier to bacteria reaching blood vessels in the lamina propria. They are not a barrier to proteins leaking out of the blood vessels. I hope this is clear as some of the issues raised below may be due to this misunderstanding.

I read the manuscript by Chang et al. Ironically, we had this manuscript cited and a whole paragraph in the introduction about what intestinal dendritic cells do that macrophage do not but we were asked to remove it as NI reviewer #1 and #2 felt it was irrelevant. We share your view that it would be nice to highlight the difference between the established function of dendritic cells to sample antigen and our new barrier function of perivascular macrophages against bacteria. As such, we have added Chang et al., along with other references back into the manuscript stating that dendritic cells could sample, take up and present antigen to lymphocytes in the intestine to affect immune responses, but Chang et al., and others have reported that there appeared to be absolutely no uptake of antigen by the macrophages. Thank you for this important suggestion this is what we added:

In fact while DCs sample antigens from the intestinal vasculature to present antigens to lymphocytes, patrol the epithelial barrier by crossing back and forth across the epithelium and migrate back to lymph nodes to present antigens, macrophages perform none of these functions²¹⁻²⁵ (Spadoni et al., Farache et al., Schulz et al., Vallon-Eberhard et al., Niess et al.).

However, in Chang et al., that was the only mention of macrophage (ie., they do not take up antigen) and no mention whatsoever that macrophage embrace blood vessels nor do they even refer to any anatomical co-location. Finally, they make no attempt to show this. I would feel very uncomfortable stating that Chang et al., showed macrophages surrounding blood vessels since there is absolutely no mention of this.

The interpretation of the data often is speculative. Some examples:

Line 147 : I don't understand how can the authors claim from the IF or flow cytometry that: 'From the flow cytometry data, it appears that it is more a loss of CCR2 and associated RFP signal than a change in CX3XR1 and a GFP signal that constitutes the red, orange, yellow and green colors.

We were asked by reviewer #3 to put this in the manuscript because the CX3CR1 only increases a tiny bit by flow cytometry whereas CCR2 drops precipitously (orders of magnitude). Please note this is the 4th iteration of this manuscript and things have been put in and pulled out to satisfy reviewers. Initially, the entire study was imaging and we did some flow cytometry to validate the conclusions and they still hold. In intravital microscopy, the ratio between the high CCR2 (Ly6C^{hi}) masks the green so the monocytes are red by imaging while on flow they appear in the upper right-hand corner (as has been shown by many). As the red disappears the cells become more orange yellow and green. By flow cytometry they also

become less red but they retain or slightly increase their GFP. This is a log scale so changes are very subtle by flow cytometry. By imaging the scale is linear. We dealt with this by calling the classical monocytes $CCR2^{hi}CX3CR1^{+}$ and the changing to macrophage as $CX3CR1^{+}CCR2^{lo}$ to indicate that the RFP signal is changing much more than the GFP signal. Nevertheless, I think it is safe to say that there is a waterfall effect of $Ly6C^{hi}$ monocytes becoming $Ly6C^{lo}$ monocytes and ultimately macrophages using imaging that has not been performed previously and validates the work of others using flow cytometry. I have added a brief statement about this flow cytometry versus imaging results. The advantage of imaging that simply could not be done with flow cytometry was the absolute documentation that only $CCR2^{hi}$ monocytes infiltrate the intestine from the vasculature. They then become less red more green with time but at no time do we see any orange or yellow or green cells adhere in the vasculature and enter into the intestines.

Line 166: Again this sentence is not supported by the data: These reporter monocytes in colonic mucosal lamina propria changed from red to orange to yellow to green, signifying a change of phenotype from $CCR2^{hi}CX3CR1^{+}$ to $CX3CR1^{+}CCR2^{lo}$ monocytes (Fig. 2a, b).. It could just be proliferation of green monocytes that have never been red. Fig 2 f, g may suggest that these cells require $CCR2$ to seed the gut, or that there is cooperation between $CCR2$ and $CX3CR1$ cells to allow the latter to seed the perivascular areas. How can the authors make sure that it is not just inability of expressing $CX3CR1$? Can they follow $CD11b+F4/80+$ cells to make sure that these cells are missing?

We have tried very hard to emphasize the value of imaging which argues against recruitment of multiple types of monocytes. While the reviewer is entirely correct that the flow cytometry cannot exclude the possibility that red and green cells arrive into the intestine and one affects the other, using imaging we never saw a single green cell ($Ly6C^{lo}CCR2^{-}$) enter the intestine from the vasculature. We now say this much more explicitly in three different places in the results. Only $CCR2^{hi}$ monocytes enter the intestine from the vasculature.

Nevertheless, we do add the following sentences to address this reviewer's concern: Although, we could only see $CCR2^{hi}CX3CR1^{+}$ and not $CX3CR1^{+}CCR2^{lo}$ monocytes entering the parenchyma from the vasculature and then converting to $CX3CR1^{+}$ monocytes and macrophages, there always exist the possibility that a small fraction of $CX3CR1^{+}CCR2^{lo}$ monocytes also migrated from the vasculature. However, adoptive transfer of $CX3CR1^{+}CCR2^{lo}$ monocytes into the vasculature does not beget any $CX3CR1^{+}$ monocytes and macrophages in the lamina propria^{8,9} arguing against this view. In addition, some of the $CX3CR1^{+}$ monocytes/macrophages could replicate after entry into the lamina propria to increase their numbers, however the literature strongly suggests there is little or no in situ proliferation by these cells⁶.

We agree with the reviewer that we cannot categorically exclude from our data that once the red monocytes become green there may be some level of replication, it is worth mentioning that numerous studies have examined this issue extensively and have concluded this does not happen. We have added this to the end of the above paragraph.

Fig. 3f. I wonder how the authors can claim this is a barrier without any functional data. The endothelium itself in the gut is organized in a barrier and includes pericytes and enteric glial cells. Functional studies have shown this (Spadoni et al. Science 2015; Mouries et al. J. Hep 2019).

Sorry if we gave the wrong impression, the macrophages do not form a barrier to vascular protein leak. If we misunderstood your question, please understand that we are in no way implying that the macrophages prevent vascular leak! The proteins are 2 nm in size and if endothelium retracts a tiny amount more proteins can leak out. The macrophages by contrast prevent bacteria from getting past them and reaching the vasculature. Bacteria are 500 nm in size and as such are caught by the macrophages and eradicated. For this we provide strong evidence. Our data using intravital microscopy, shows that the majority of the bacteria that enter the parenchyma are taken up by the perivascular macrophages keeping them away from the vasculature. Using intravital imaging we can clearly show the uptake of bacteria by CX3CR1+ macrophages in living mice. NI reviewer #2 asked us to show this is specifically macrophages and not dendritic cells or other cells (endothelium) and specific depletion of these macrophages allows for more dissemination of bacteria into the blood stream. We realize that the above papers showed that when there is increased endothelial permeability, the bacteria cross into the vasculature more easily. Nevertheless, there is no reason why the two systems could not co-exist. The papers you provide specifically targeted molecules in endothelium to show bacteria cannot cross into circulation ie reach the liver. When we affect the macrophages, we see increased bacterial numbers reach the liver ie., similar results suggesting both endothelium and macrophages function as a barrier for bacteria. Based on your suggestions we add the following:

It is worth mentioning that these lamina propria macrophages are only one of numerous barriers to bacterial entry from the bowel lumen into the bloodstream. It has been known for a long time that the epithelium, IgA and the mucus in the gut all function as excellent barriers to most bacteria³⁷. However, *S. aureus* primarily in newborns²⁹ and *S. typhimurium* in all humans can bypass the mucosal barrier and enter the parenchyma. Spadoni et al., and Mouries et al., reported that *S. typhimurium* but not *E. coli* could disrupt the endothelial barrier to translocate into the blood stream^{21, 38} suggesting blood vessels themselves were a microbial barrier. Our work adds a third barrier between the epithelium and endothelium, a potent immunological barrier that prevents the bacteria from getting to the endothelial barrier after penetrating the epithelium. These macrophages unlike endothelium or other vessel-associated cells, played the unique function of phagocytosing these pathogenic bacteria that crossed the mucosal barrier. Indeed, we found bacteria inside the macrophages surrounding the vessels but not inside endothelium or other vessel-associated cells. Lack of microbiota, resulted in gaps between these perivascular macrophages, and now bacteria could be seen adhering and crossing the vessel wall and entering into the bloodstream. Similarly, Mouries et al., reported that an altered microbiota due to high fat diet was capable of disrupting the endothelial barrier³⁸ suggesting that both the vascular and macrophage barriers may be affected by microbiota. Our data do not rule out the possibility that mucosal and endothelial barriers in addition to the perivascular macrophages are also necessary to prevent translocation of bacteria into the bloodstream.

Fig. 3g. The vasculature has been shown to be underdeveloped in germ free mice (Reinhardt C, et al. Nature 2012). Also in this image the vasculature seems not properly developed.

We agree with the reviewer that the image provided in 3g is a terrible example. This is due to the threshold settings from the image analysis and the fact that the tissue is not perfectly flat making it look like there are discontinuities which would mean overt bleeding. This is not the case as we are imaging the area and see no overt bleeding. We have now included a more representative image of the

vasculature in this Figure using 3D reconstruction clearly showing intact vasculature. In addition, we have used imaging software (to avoid bias) to calculate how much vasculature is in each of the two strains of mice and the amount of vasculature in SPF and GF mice is identical. I would like to mention that the authors of the aforementioned paper clearly state that they only saw this less dense vasculature in the mid-distal part of the small intestine and not elsewhere and the difference was about 25%. Our data are from the colon. We now cite this paper and add the following to draw attention to this important matter.

Although germ-free mice have been reported to have less dense vasculature in the mid-distal part of the small intestine²⁸, in the colon of our germ free mice and SPF mice, they had identical intestinal vasculature density so that alterations in vasculature could not account for the macrophage distribution difference (Fig. 3g). What was striking was the clear adhesion of the macrophages to the lamina propria microcirculation, but the spectacular retraction of the perivascular macrophages exposing large areas of the vasculature in the absence of a microbiota (Fig. 3f, g).

Fig. 5. The experiments of pathogen dissemination in antibiotic treated mice may be explained by several other possibilities:

1. Mice lacking the microbiota are more affected by pathogens because the microbiota releases antimicrobial peptides

We totally agree with the reviewer that antibiotic treated mice have lost their pathogen colonization resistance by removing the microbiota, and we have now added this in our manuscript (see below). However, we used a high inoculum to get large numbers of bacteria into the colon parenchyma. We clearly see the uptake of the bacteria by the perivascular macrophages during these infections in untreated animals, but we see a very impressive leak of bacteria into the blood vessels after antibiotic treatment. Please remember that if we remove macrophages without affecting the microbiota than we also see increased dissemination- clearly the macrophages are important in limiting dissemination.

It is worth mentioning that these lamina propria macrophages are only one of numerous barriers to bacterial entry from the bowel lumen into the bloodstream. It has been known for a long time that the commensal bacteria, epithelium, IgA and the mucus in the gut all function as excellent barriers to most bacteria³⁷. However, *S. aureus* primarily in newborns²⁹ and *S. typhimurium* in all humans can bypass the mucosal barrier and enter the parenchyma.

2. The microbiota induces IgA responses. IgAs quickly go down in the absence of the microbiota and hence also this protection is lost.

3. The microbiota induces mucus production and hence also this barrier is lost.

These are great points and we have added this to a section of our discussion (highlighted in yellow above). It's worth noting that neither IgA nor the mucus completely prevent Salmonella from disseminating and if this pathogen gets into the parenchyma it readily crosses into the vasculature in antibiotic treated mice and is hugely reduced in number in untreated SPF mice due to the macrophages that are surrounding the vasculature. Similarly, because we are imaging, we can watch the bacteria in

the parenchyma after they have crossed the two barriers you mention and in microbiota depleted mice the bacteria reach the vasculature and in the untreated mice they are mostly in intestinal macrophages.

4. Antibiotic treatment impairs the formation of a vascular barrier and thus this is more permeable to bacteria (Mouries et al. J Hep 2019).

Thank you for this comment. We have now cited this paper and add the paragraph below.

Spadoni et al., and Mouries et al., reported that *S. typhimurium* but not *E. coli* could disrupt the endothelial barrier to translocate into the blood stream^{21, 38} suggesting blood vessels themselves were a microbial barrier. Our work adds a third barrier between the epithelium and endothelium, a potent immunological barrier that prevents the bacteria from getting to the endothelial barrier after penetrating the epithelium. These macrophages unlike endothelium or other vessel-associated cells, played the unique function of phagocytosing these pathogenic bacteria that crossed the mucosal barrier. Indeed, we found bacteria inside the macrophages surrounding the vessels but not inside endothelium or other vessel-associated cells. Lack of microbiota, resulted in gaps between these perivascular macrophages, and now bacteria could be seen adhering and crossing the vessel wall and entering into the bloodstream. Similarly, Mouries et al., reported that an altered microbiota due to high fat diet was capable of disrupting the endothelial barrier³⁸ suggesting that both the vascular and macrophage barriers may be affected by microbiota. Our data do not rule out the possibility that mucosal and endothelial barriers in addition to the perivascular macrophages are also necessary to prevent translocation of bacteria into the bloodstream.

That macrophages readily internalize bacteria is a known function.

Absolutely and we state this. We are certainly not claiming that macrophage uptake of bacteria is novel. The perivascular localization of the macrophages to prevent bacteria from crossing into the blood stream and the impact on macrophages by the microbiota is novel.

Fig 5K. The increased dissemination in mice lacking macrophages is quite expected as they play a major role anyway in microbial elimination. I am quite surprised that there was no difference in the gut even though the authors analyzed only the colon which is not the best site for Salmonella infection (they should have analyzed also the small intestine).

The purpose of the macrophage depletion experiment was because NI reviewer #2 stated that we could not claim that dendritic cells were not the critical cell involved in this bacterial uptake. Similarly, your comment that it could all be endothelial cells that form the barrier not macrophages is also important. As such, we went back to the laboratory and generated *Csf1^{LSL-DTR}/LysM^{cre}* mice which permitted selective depletion of macrophages, but not dendritic cells or other cell types in the intestines. We saw a decrease in dissemination of bacteria into the blood stream in these macrophage depleted mice. This is what allows us to conclude that the macrophages are important as a barrier to bacteria reaching the vasculature and are critical for addressing 3 of the points you raise above about what evidence do we have that the macrophage function as a barrier to bacteria. The Salmonella experiments into the colon were requested by NI reviewers #1 and #2 mainly because of what we saw with *S. aureus* in the colon and the fact that much of our work was in the colon. Salmonella was used as a model organism to show it too could penetrate the macrophage barrier in the colon.

Fig. 6a. Probably the most convincing experiment of monocyte recruitment was done with thermal ablation. Here it is clear that CCR2 are recruited and probably transform into CX3CR1 (even though the clear demonstration of a transformation of these cells from CCR2+ to CX3CR1+ cells would come using a tracer system like ROSA26. Again the authors cannot exclude that CCR2+ cells favor the recruitment of CX3CR1+ cells. Indeed in CCR2-/- mice CX3CR1+ cells eventually seed the perivascular region, suggesting that they can arrive there even in the absence of CCR2.

We have never seen any influx of CX3CR1 cells into the intestines that are not positive for CCR2 in any of our systems and we state this very clearly. We also very clearly state that CCR2-/- mice were dramatically delayed in their recruitment of monocytes and their transition to macrophages. Indeed, we also very clearly state that eventually some red monocytes do make it to the site and this can be abolished by removing the microbiota which is the other factor that recruits CCR2 monocytes. Because we can image we can categorically state that red CCR2 positive monocytes always preceded yellow cells which then became green cells and at no point did we see the latter enter from the vasculature. In addition, in the bone marrow transfer experiments, the recipient has no green cells in the intestine except in bone marrow and bone marrow derived green cells never appeared in the intestine without first being red inflammatory monocytes. Nevertheless, as we indicated above, we have added the paragraph about the possibility that some minor amount of CX3CR1 monocytes being recruited even though we do not see it.

Although, we could only see CCR2^{hi}CX3CR1⁺ and not CX3CR1⁺CCR2^{lo} monocytes entering the parenchyma from the vasculature and then converting to CX3CR1⁺ monocytes and macrophages, there always exist the possibility that a small fraction of CX3CR1⁺CCR2^{lo} monocytes also migrated from the vasculature. However, adoptive transfer of CX3CR1⁺CCR2^{lo} monocytes into the vasculature does not begot any CX3CR1⁺ monocytes and macrophages in the lamina propria^{8,9} arguing against this view.

Fig. 8. The experiments of DSS in abx treated mice are somehow misleading as germ free mice succumb because they have defects in the epithelial barrier and not because of higher inflammation (Hernández-Chirilaque C et al J Crohns Colitis. 2016) Indeed they lack the bacteria responsible for the induction of the inflammatory response in the presence of DSS. This may have nothing to do with the macrophages lining the vessels.

We thank the reviewer for this important point. I read the paper you referred, and we see nearly identical results. The cytokine storm is diminished dramatically including the CCR2 ligand in antibiotic treated mice. Our new data suggest that monocyte recruitment is dramatically reduced leading to the intestines and cecum being full of blood due to poor repair. In addition, the NR4a1-/- mice which had delayed conversion of monocytes only had a minor phenotype as DSS colitis was hypothesized to delay monocyte conversion. As such we do not disagree that there is a very profound breach of the epithelium, but there is also injury to the vasculature and the healing of the vasculature is impaired in the microbiota depleted mice and as such the profound lack of healing can be seen. Again, a simple disruption of epithelium is not the only defect in DSS, and our results can co-exist and extend the previous DSS results. Nevertheless, we have cited the paper suggested by the reviewer and highlight the additional new data.

Our data also revealed an increased impairment when DSS treated mice received Abx, but the underlying mechanism was not increased inflammation. Indeed, the shortening of the colon (Fig. 8a), used as a marker of inflammatory injury as well as all of the pro-inflammatory cytokines and chemokines

were lower in the Abx-treated mice that received DSS than in DSS treated mice that received no Abx (Fig. 8b,c) confirming previous results³¹ (Hernández-Chirilaque C et al J Crohns Colitis. 2016). Moreover, Abx-treated mice had reduced induction of MCP-1 (CCL2), the key chemokine that attracts CCR2⁺ monocytes, in the colon of DSS treated mice (Fig. 8c). In addition, TNF- α and IL-6, cytokines known to be mainly produced by macrophages and promote regeneration of damaged intestinal mucosa by acting on epithelial cells²⁷ were reduced by 90% (Fig. 8b). Next, when we imaged the lamina propria of control and Abx-treated *Cx3cr1*^{GFP/+} *Ccr2*^{RFP/+} mice 7 days after the start of DSS there was tremendous accumulation of CCR2⁺ and CX3CR1⁺ cells in control mice (Fig. 8d). By contrast, Abx-treated mice showed less accumulation of these cells, implying a delay of CX3CR1⁺ monocytes/macrophages appearance within areas of injury. The reconstruction of blood vessels at the injured mucosal lesion was also inhibited and showed higher permeability, reflecting less mature vasculature in Abx-treated mice (Fig. 8e). Flow cytometry data confirmed these imaging results; both CD80⁺CD206⁻ and CD80⁻CD206⁺ CX3CR1^{hi} macrophages were decreased in Abx-treated mice (Fig. 8f). Indeed, 7 days after the start of 2% DSS water, all Abx-treated mice demonstrated overt sickness behaviour and all of them died at approximately 9 days (Fig. 8g). There was a very large amount of frank hemorrhage found throughout the intestinal tract when DSS colitis mice received antibiotics (Fig. 8h). This very likely contributed to the reduced survival. This is consistent with the mucosal lamina propria CX3CR1⁺ macrophages helping in the healing process and in particular in the restitution of a functional vasculature. In the absence of the microbiome, fewer monocytes enter the injury in a timely fashion and repair is delayed.

REVIEWERS' COMMENTS:

Reviewer #4 (Remarks to the Author):

I wanted to compliment the authors for clearly respond to my comments and to convince me about their points.

I have only one minor point: the authors could hypothesize in their discussion that macrophages may participate in the generation of the gut vascular barrier and thus may both perform a direct barrier function and an indirect by favoring the development of the gut vascular barrier.

Response to the Reviewer #4

I wanted to compliment the authors for clearly respond to my comments and to convince me about their points.

I have only one minor point: the authors could hypothesize in their discussion that macrophages may participate in the generation of the gut vascular barrier and thus may both perform a direct barrier function and an indirect by favoring the development of the gut vascular barrier.

First we thank reviewer 4 who stated “I wanted to compliment the authors for clearly respond to my comments and to convince me about their points.” I have only one minor point: the authors could hypothesize in their discussion that macrophages may participate in the generation of the gut vascular barrier and thus may both perform a direct barrier function and an indirect by favoring the development of the gut vascular barrier. We have added a sentence about the possibility of macrophages regulating permeability indirectly and directly. “It is also possible that the perivascular macrophages which have direct contact with the blood vessels could regulate and increase the vascular barrier function. Indeed permeability was greatly increased in new blood vessels during repair when perivascular macrophages were lacking”.